# Formalizing Generalization and Adversarial Robustness of Neural Networks to Weight Perturbations

**Yu-Lin Tsai**
National Yang Ming Chiao Tung University
uriah1001@gmail.com*

**Chia-Yi Hsu**
National Yang Ming Chiao Tung University
chiayihsu8315@gmail.com

**Chia-Mu Yu**
National Yang Ming Chiao Tung University
chiamuyu@gmail.com

**Pin-Yu Chen**
IBM Research
pin-yu.chen@ibm.com

## Abstract

Studying the sensitivity of weight perturbation in neural networks and its impacts on model performance, including generalization and robustness, is an active research topic due to its implications on a wide range of machine learning tasks such as model compression, generalization gap assessment, and adversarial attacks. In this paper, we provide the first integral study and analysis for feed-forward neural networks in terms of the robustness in pairwise class margin and its generalization behavior under weight perturbation. We further design a new theory-driven loss function for training generalizable and robust neural networks against weight perturbations. Empirical experiments are conducted to validate our theoretical analysis. Our results offer fundamental insights for characterizing the generalization and robustness of neural networks against weight perturbations.

## 1 Introduction

Neural network is currently the state-of-the-art machine learning model in a variety of tasks, including computer vision, natural language processing, and game-playing, to name a few. In particular, feed-forward neural networks consists of layers of trainable model weights and activation functions with the premise of learning informative data representations and the complex mapping between data samples and the associated labels. Albeit attaining superior performance, the need for studying the sensitivity of neural networks to weight perturbations is also intensifying owing to several practical motivations. For instance, in model compression, the robustness to weight quantization is crucial for designing energy-efficient hardware accelerator [20] and for reducing memory storage while retaining model performance [9, 26]. The notion of weight perturbation sensitivity is also used as a property to reflect the generalization gap at local minima [10, 15]. In adversarial robustness and security, weight sensitivity can be leveraged as a vulnerability for fault injection and causing erroneous prediction [11, 32]. However, while weight sensitivity plays an important role in many machine learning tasks and problem setups, theoretical characterization of its impacts on generalization and robustness of neural networks remains elusive.

This paper bridges this gap by developing a novel theoretical framework for understanding the generalization gap (through Rademacher complexity) and the robustness (through classification margin) of neural networks against norm-bounded weight perturbations. Specifically, we consider

---

[1] Work partly done in TCFSH

35th Conference on Neural Information Processing Systems (NeurIPS 2021).

the multi-class classification problem setup and multi-layer feed-forward neural networks with non-negative monotonic activation functions. Our analysis offers fundamental insights into how weight perturbation affects the generalization gap and the pairwise class margin. To the best of our knowledge, this study is the first work that provides a comprehensive theoretical characterization of the interplay between weight perturbation, robustness in classification margin, and generalization gap. Moreover, based on our analysis, we propose a theory-driven loss function for training generalizable and robust neural networks against norm-bounded weight perturbations. We validate its effectiveness via empirical experiments. Our main contributions are summarized as follows.

- We study the robustness (worst-case bound) of the pairwise class margin function against weight perturbations in neural networks, including the analysis of single-layer (Theorem 1), all-layer (Theorem 2), and selected-layer (Theorem 3) weight perturbations.
- We characterize the generalization behavior of robust surrogate loss for neural networks under weight perturbations (Section 3.4) through Rademacher complexity (Theorem 4).
- We propose a theory-driven loss design for training generalizable and robust neural networks (Section 3.5). The empirical results in Section 4 validate our theoretical analysis and demonstrate the effectiveness of improving generalization and robustness against weight perturbations. We also show that in our studied setting the associated generalization bounds are non-vacuous.

## 2 Related Works

### 2.1 Generalization in Standard Settings

In model compression, the robustness to weight quantization is critical to reducing memory size and accesses for low-precision inference and training [9]. [20] showed that neural models that are robust to random bit flipping errors are beneficial for operating in low-voltage regime and thus improving energy efficiency for hardware implementation. [26] showed that incorporating weight perturbation sensitivity into training can better retain model performance (standard accuracy) after quantization. For studying the generalization of neural networks, [10] proposed a metric called sharpness (or weight sensitivity) by perturbing the learned model weights around the local minima of the loss landscape for generalization assessment. Similar concepts of considering nearby parameter loss is as well presented in [18] where this concept is to circumvent from falling into sharp empirical loss pit with high generalization gap. [1] introduced weight noise into the training process and concluded that random noise training improves the overall generalization. [27] empirically showed that using adversarial weight perturbation during training can improve generalization against input perturbations. [5] obtained non-vacuous generalization bound on deep (stochastic) neural networks by directly optimizing the PAC-Bayes bound while [16] as well combined weight perturbation with PAC-Bayes analysis to derive the generalization bound. [15] made a connection between sharpness and PAC-Bayes theory and found that some combination of sharpness and norms on the model weights may capture the generalization behavior of neural networks. Additionally, [4] discovered normalized margin measure to be useful towards quantifying generalization property and a bound was therefore constructed to give an quantitative description on the generalization gap. Moreover, [7] incorporated additional assumptions to offer tighter and size-independent bounds from the setting of [17] and [4] respectively. We note that previous methods use random weight perturbations to evaluate the generalization ability (e.g. sharpness), which we call a standard generalization setting. In contrast, we study a different problem setup: generalization of models trained under worst-case (adversarial) weight perturbation.

Despite development of various generalization bounds, empirical observations in [14] showed that once the size of the training dataset grows, generalization bounds proposed in [15] and [4] will enlarge and thus become vacuous. Discussions on the relation between [14] and our works can be found at Appendix E, where we show that in our studied setting the associated generalization bounds are non-vacuous. On the other hand, [3] and [22] applied several techniques in tandem with a probabilistic method named path sampling to construct a representing set of given neural networks for approximating and studying the generalization property. Another approach considered by [19] consists of segmenting the neural networks into two functions, predictor and feature selection respectively, where two measures (representativeness and feature robustness) concerning these aforementioned functions were later combined to offer a meaningful generalization bound. However, these works only focused on the generalization behavior of the local minima and did not consider

the generalization and robustness under weight perturbations. [26] proposed a certification method for weight perturbation retaining consistent model prediction. While the certification bound can be used to train robust models with interval bound propagation [8], it requires additional optimization subroutine and computation costs when comparing to our approach. Moreover, the convoluted nature of certification bound complicates the analysis when studying generalization, whereas our theoretical characterization is an intuitive and tight worst-case analysis.

## 2.2 Adversarial Robustness and Input/Weight Perturbation

In adversarial robustness, fault-injection attacks are known to inject errors to model weights at the inference phase and causing erroneous model prediction [11, 32], which can be realized at the hardware level by changing or flipping the logic values of the corresponding bits and thus modifying the model parameters saved in memory [2, 24]. [31] proposed to use the mode connectivity of the model parameters in the loss landscape for mitigating such weight-perturbation-based adversarial attacks.

Although, to the best of our knowledge, theoretical characterization of generalization and robustness for neural networks against *weight* perturbations remains elusive, recent works have studied these properties under another scenario — *input* perturbations. Both empirical and theoretical evidence suggests the existence of a fundamental trade-off between generalization and robustness against norm-bounded input perturbations [28, 21, 30, 23]. The adversarial training proposed in [12] is a popular training strategy for training robust models against input perturbations, where a min-max optimization principle is used to minimize the worst-case input perturbations of a data batch during model parameter updates. For adversarial training with input perturbations, [25] proved its convergence and [29] derived bounds on its Rademacher complexity for generalization. Different from the case of input perturbation, we note that min-max optimization on neural network training subject to weight perturbation is not straightforward, as the minimization and maximization steps are both taken on the model parameters. In this paper, we disentangle the min-max formulation for weight perturbation by developing bounds for the inner maximization step and provide quantifiable metrics for training generalizable and robust neural networks against weight perturbations.

## 2.3 Comparison against Related Generalization Bound

In the following subsection below we will provide the discussion to our characterization of generalization bound against similar notions in other related works. For the convenience of comparison, we formulated it as a table in Appendix H. In what follows we offer the detailed comparison on different settings, equation setup between ours and these related notions.

In [6], the authors extended the result of sharpness [15] and integrated this concept as a part of the training process, namely the sharpness-aware minimization. In the paper, the author instead of directly solving the inner maximization loop, uses linear approximation to resolve the issue. Furthermore, the generalization bound given in [6] considers only the standard setting, in which the generalization gap is measured between population risk and the sharpness-aware sample risk, differing from our adversarial setting where the generalization gap is contrast between both adversarial population and sample risk.

On the other hand, the authors in [27] propose to utilize weight perturbation to mitigate the widening robust generalization gap (based on input robustness). In [27], the author trains the model by injecting the so-called "double perturbation" into the training process. We note that this is different from our approach of constructing adversarial robustness on weight space. Specifically, the weight perturbation in [27] (second maximization step in equation (7), [27]) is determined after every corresponding adversarial example is calculated, while our approach focuses on the worst-case weight perturbation given every sample data. Moreover, the generalization gap extends from [15] where it is measured against the population risk and the sample risk along with perturbation direction sampled from zero mean spherical Gaussian distribution. On the contrary, our setting considers a generalization gap from both the adversarial population risk and sample risk.

# 3 Main Results

We provide an overview of the presentation flow for our main results as follows. First, we introduce the mathematical notations and preliminary information in Section 3.1. In Section 3.2, we establish our weight perturbation analysis on a simplified case of single-layer perturbation. We then use the single-layer analysis as a building block and extend the results to the multi-layer perturbation setting in Section 3.3. Here we would like to note that similar reasoning could be applied to obtain results in convolutional neural networks and we relay the corresponding results and experiments in Appendix A.4, respectively. In Section 3.4, we define the framework of robust training with surrogate loss and study the generalization property using Rademacher complexity. Finally, we propose a theory-driven loss toward training robust and generalizable neural networks in Section 3.5.

## 3.1 Notation and Preliminaries

**Notation**  We start by introducing the mathematical notations used in this paper. We define the set $[L] := \{1, 2, ..., L\}$. For any two non-empty sets $A$, $B$, $\mathbb{F}_{A \mapsto B}$ denotes the set of all functions from $A$ to $B$. We mark the indicator function of an event E as $\mathbb{1}(E)$, which is 1 if $E$ holds and 0 otherwise. We use $sgn(\cdot)$ to denote element-wise sign function that outputs 1 when input is nonnegative and -1 otherwise. Boldface lowercase letters are used to denote vectors (e.g., $\mathbf{x}$), and the $i$-th element is denoted as $[\mathbf{x}]_i$. Matrices are presented as boldface uppercase letters, say $\mathbf{W}$. Given a matrix $\mathbf{W} \in \mathbb{R}^{k \times d}$, we write its $i$-th row, $j$-th column and $(i, j)$ element as $W_{i,:}$, $W_{:,j}$, and $W_{i,j}$ respectively. Moreover, we write its transpose matrix as $(\mathbf{W})^T$. The matrix $(p, q)$ norm is defined as $\|\mathbf{W}\|_{p,q} := \left\| \left[ \|W_{:,1}\|_p, \|W_{:,2}\|_p, ..., \|W_{:,d}\|_p \right] \right\|_q$ for any $p, q \geq 1$. For convenience, we have $\|\mathbf{W}\|_p = \|\mathbf{W}\|_{p,p}$ and write the spectral norm and Frobenius norm as $\|\mathbf{W}\|_\sigma$ and $\|\mathbf{W}\|_F$ respectively. We mark one matrix norm commonly used in this paper – the matrix $(1, \infty)$ norm. With a matrix $\mathbf{W}$, we express its matrix $(1, \infty)$ norm as $\|\mathbf{W}\|_{1,\infty}$, which is defined as $\|\mathbf{W}\|_{1,\infty} = \max_j \|W_{:,j}\|_1$ and $\|\mathbf{W}^T\|_{1,\infty} = \max_i \|W_{i,:}\|_1$. We use $\mathbb{B}_{\mathbf{W}}^\infty(\epsilon)$ to express an element-wise $\ell_\infty$ norm ball of matrix $\mathbf{W}$ within radius $\epsilon$, i.e., $\mathbb{B}_{\mathbf{W}}^\infty(\epsilon) = \{\hat{\mathbf{W}} \mid |\hat{W}_{i,j} - W_{i,j}| \leq \epsilon, \forall i \in [k], j \in [d]\}$.

**Preliminaries**  In order to formally explain our theoretical results, we introduce the considered learning problem, neural network model, and complexity definition. Let $\mathcal{X}$ and $\mathcal{Y}$ be the feature space and label space, respectively. We place the assumption that all data are drawn from an unknown distribution $\mathcal{D}$ over $\mathcal{X} \times \mathcal{Y}$ and each data point is generated under i.i.d condition. In this paper, we specifically consider the feature space $\mathcal{X}$ as a subset of $d$-dimensional Euclidean space, i.e., $\mathcal{X} \subseteq \mathbb{R}^d$. We denote the symbol $\mathcal{F} \subseteq \mathbb{F}_{\mathcal{X} \mapsto \mathcal{Y}}$ to be the hypothesis class which we use to make predictions. Furthermore, we consider a loss function $\ell : \mathcal{X} \times \mathcal{Y} \longrightarrow [0, 1]$ and compose it with the hypothesis class to make a function family written as $\ell_\mathcal{F} := \{(\mathbf{x}, y) \mapsto \ell(f(\mathbf{x}), y) \mid f \in \mathcal{F}\}$. The optimal solution of this learning problem is a function $f^* \in \mathcal{F}$ such that it minimizes the population risk $R(f) = E_{(\mathbf{x},y) \sim \mathcal{D}}[\ell(f(\mathbf{x}), y)]$. However, since the underlying data distribution is generally unknown, one typically aims at reducing the empirical risk evaluated by a set of training data $\{(\mathbf{x}_i, y_i)\}_{i=1}^n$, which can be expressed as $R_n(f) = \frac{1}{n} \sum_{i=1}^n \ell(f(\mathbf{x}_i), y_i)$. The generalization error is the gap between population and empirical risk, which could serve as an indicator of model's performance under unseen data from identical distribution $\mathcal{D}$.

To study the generalization error, one would explore the learning capacity of a certain hypothesis class. In this paper, we adopt the notion of Rademacher complexity as a measure of learning capacity, which is widely used in statistical machine learning literature [13]. The empirical Rademacher complexity of a function class $\mathcal{F}$ given a set of samples $\mathcal{S} = \{(\mathbf{x}_i, y_i)\}_{i=1}^n$ is $\mathcal{R}_\mathcal{S}(\ell_\mathcal{F}) = E_\nu[\sup_{f \in \mathcal{F}} \frac{1}{n} \sum_{i=1}^n \nu_i \ell(f(\mathbf{x}_i), y_i)]$ where $\{\nu_i\}_{i=1}^n$ is a set of i.i.d Rademacher random variables with $\mathbb{P}\{\nu_i = -1\} = \mathbb{P}\{\nu_i = +1\} = \frac{1}{2}$. The empirical Rademacher complexity measures on average how well a function class $\mathcal{F}$ correlates with random noises on dataset $\mathcal{S}$. Thus, a richer or more complex family could better correlate with random noise on average. With Rademacher complexity as a toolkit, one can develop the following relationship between generalization error and complexity measure. Specifically, it is shown in [13] that given a set of training samples $\mathcal{S}$ and assume that the range of loss function $\ell(f(\mathbf{x}), y)$ is $[0, 1]$. Then for any $\delta \in (0, 1)$, with at least probability $1 - \delta$ we have $\forall f \in \mathcal{F}, R(f) \leq R_n(f) + 2\mathcal{R}_\mathcal{S}(\ell_\mathcal{F}) + 3\sqrt{\frac{\log \frac{2}{\delta}}{2n}}$ Note that when the

Rademacher complexity is small, it is then viable to learn the hypothesis class $\mathcal{F}$ by minimizing the empirical risk and thus effectively reducing the generalization gap.

Finally, we define the structure of neural networks and introduce a few related quantities. The problem studied in this paper is a multi-class classification task with the number of classes being $K$. Consider an input vector $\mathbf{x} \in \mathcal{X} \subseteq \mathbb{R}^d$, an L-layer neural network is defined as $f_{\boldsymbol{W}}(\mathbf{x}) = \mathbf{W}^L(...\rho(\mathbf{W}^1\mathbf{x})...) \in \mathbb{F}_{\mathcal{X} \mapsto \mathbb{R}}$ with $\boldsymbol{W}$ being the set containing all weight matrices, i.e., $\boldsymbol{W} := \{\mathbf{W}^k | \forall k \in [L]\}$, and the notation $\rho(\cdot)$ is used to express any non-negative monotone activation function and we further assume that $\rho(\cdot)$ is 1-Lipschitz, which includes popular activation functions such as ReLU applied element-wise on a vector. Moreover, the $i$-th component of neural networks' output is written as $f_{\boldsymbol{W}}^i(\mathbf{x}) = [f_{\boldsymbol{W}}(\mathbf{x})]_i$ and a pairwise margin between $i$-th and $j$-th class, denoted as $f_{\boldsymbol{W}}^{ij}(\mathbf{x}) := f_{\boldsymbol{W}}^j(\mathbf{x}) - f_{\boldsymbol{W}}^j(\mathbf{x})$, is said to be the difference between two classes in output of the neural network. Lastly, we use the notion of $\mathbf{z}^k$ and $\hat{\mathbf{z}}^k$ to represent the output vector of the $k$-th layer ($k \in [L-1]$) under natural and weight perturbed settings respectively, which are $\mathbf{z}^k = \rho(\mathbf{W}^k(...\rho(\mathbf{W}^1\mathbf{x})...))$ and $\hat{\mathbf{z}}^k = \rho(\hat{\mathbf{W}}^k(...\rho(\hat{\mathbf{W}}^1\mathbf{x})...))$, where $\hat{\mathbf{W}}^i \in \mathbb{B}_{\mathbf{W}^i}^\infty(\epsilon_i)$ denotes the perturbed weight matrix bounded by its element-wise $\ell_\infty$-norm with radius $\epsilon_i$ for some $i \in [k]$. Throughout this paper, we study the weight perturbation setting that each layer has an independent layer-wise perturbation budget, denoted by $\epsilon_k, \forall k \in [L]$. In the case of single-layer perturbation, we may omit the layer index and simply use $\epsilon$.

## 3.2 Building Block: Single-Layer Weight Perturbation

We study the sensitivity of neural network to weight perturbations through the pairwise margin bound $f_{\boldsymbol{W}}^{ij}(\mathbf{x})$. Specifically, when $i$ and $j$ corresponds to the top-1 and the second-top class prediction of $\mathbf{x}$, respectively, the margin can be used as an indicator of robust prediction under weight perturbation to $\boldsymbol{W}$. For ease of understanding, we have provided a simple example of a three-layer neural network and explain the bound through the error propagation incurred by weight perturbation in Appendix A.1. Here we directly introduce and analyze our theorem below.

**Theorem 1 (N-th layer weight perturbation ($N \neq L$))** *Let $f_{\boldsymbol{W}}(\boldsymbol{x}) = \boldsymbol{W}^L(...\rho(\boldsymbol{W}^1\boldsymbol{x})...)$ denote an L-layer neural network and let $f_{\widehat{\boldsymbol{W}}}(\boldsymbol{x}) = \boldsymbol{W}^L(..\hat{\boldsymbol{W}}^N...\rho(\boldsymbol{W}^1\boldsymbol{x})...)$ with $\hat{\boldsymbol{W}}^N \in \mathbb{B}_{\boldsymbol{W}^N}^\infty(\epsilon_N), N \neq L$, denote the corresponding network subject to N-th layer perturbation. For any set of perturbed and unperturbed pairwise margin $f_{\widehat{\boldsymbol{W}}}^{ij}(\boldsymbol{x})$ and $f_{\boldsymbol{W}}^{ij}(\boldsymbol{x})$, we have*

$$f_{\widehat{\boldsymbol{W}}}^{ij}(\boldsymbol{x}) \leq f_{\boldsymbol{W}}^{ij}(\boldsymbol{x}) + \epsilon_N \left\| W_{i,:}^L - W_{j,:}^L \right\|_1 \left\| \boldsymbol{z}^{N-1} \right\|_1 \Pi_{k=1}^{L-N-1} \left\| (\boldsymbol{W}^{L-k})^T \right\|_{1,\infty}$$

$$:= f_{\boldsymbol{W}}^{ij}(\boldsymbol{x}) + \Delta(\epsilon_N; \boldsymbol{z}^{N-1}; f)$$

*where $\boldsymbol{z}^k = \rho(\boldsymbol{W}^k(...\rho(\boldsymbol{W}^1\boldsymbol{x})...))$.*

*Proof*: See Appendix A.2

Since the final layer does not have any activation function, the margin bound on the margin difference when only perturbing the final layer can be simply derived, which is given in the following lemma.

**Lemma 1 (Final-layer weight perturbation)** *Consider the case $N = L$ in Theorem 1, we have*

$$f_{\widehat{\boldsymbol{W}}}^{ij}(\boldsymbol{x}) \leq f_{\boldsymbol{W}}^{ij}(\boldsymbol{x}) + 2\epsilon_L \left\| \boldsymbol{z}^{L-1} \right\|_1$$

*where $\boldsymbol{z}^{L-1}$ is the output of the $(L-1)$-th layer.*

*Proof*: See Appendix A.2.

## 3.3 General Setting: Multi-Layer Weight Perturbation

With the developed single-layer analysis in Section 3.2 as a building block, we now extend our analysis to the general setting of multi-layer weight perturbation, which is further divided into two cases: (i) the case of perturbing all $L$ layers; and (ii) the case of perturbing $I$ out of $L$ layers.

### 3.3.1 Perturbing All Layers

Once equipped with the concept of error propagation over subsequent layers, we consider the scenario where every layer in a neural network is subject to weight perturbation.

For simplicity and understanding, we will introduce an notation previously used but not fully explained in Theorem 1 and Lemma 1. Specifically, the notion $\Delta(\cdot)$ can be described as the error caused by certain layer. Moreover, as we will see later, the function requires three inputs, the perturbation of that certain layer, the previous input and finally the neural network. Explicitly, for an $L$-layer neural network $f(\cdot)$, a perturbation radius of the $k$-th layer $\epsilon_k$, and a prior input $\mathbf{z}$, the delta function can be described as the following,

$$\Delta(\epsilon_k; \mathbf{z}; f) = \epsilon_k \left\| W_{i,:}^L - W_{j,:}^L \right\|_1 \|\mathbf{z}\|_1 (\Pi_{m=k+1}^{L-1} \|\mathbf{W}^m\|_{1,\infty})$$

We denote the model under this case as the all-perturbed setting. The following theorem states an upper bound on the pairwise margin between the natural (unperturbed) and all-perturbed settings.

**Theorem 2 (all-layer perturbation)** *Let* $f_{\mathbf{W}}(x) = W^L(...\rho(W^1 x)...)$ *denote an L-layer (natural) neural network and let* $f_{\widehat{\mathbf{W}}}(x) = \hat{W}^L(..\hat{W}^N...\rho(\hat{W}^1 x)...)$ *with* $\hat{W}^k \in \mathbb{B}_{\mathbf{W}^k}^\infty(\epsilon_k),\ \forall k \in [L]$, *denote its perturbed version. For any set of pairwise margin* $f_{\widehat{\mathbf{W}}}^{ij}(x)$ *and* $f_{\mathbf{W}}^{ij}(x)$, *we have*

$$f_{\widehat{\mathbf{W}}}^{ij}(x) \le f_{\mathbf{W}}^{ij}(x) + \underbrace{\Delta(\epsilon_1; \mathbf{x}; f)}_{\textit{Input Layer Error}} + \underbrace{\sum_{k=2}^{L-1} \Delta(\epsilon_k; z^{k-1^*}; f)}_{\textit{Intermediate Layer Error}} + \underbrace{2\epsilon_L \left\| z^{L-1^*} \right\|_1}_{\textit{Final Layer Error}}$$

*where* $z^{k^*} = \rho(\mathbf{W}^{k^*}...\rho(\mathbf{W}^{1^*}x))$ *with* $\mathbf{W}^{k^*}$ *defined as*

$$\begin{cases} W_{i,j}^{k^*} = W_{i,j}^k + \epsilon_k,\ \forall i, j \text{ and } \forall k \in [L] \setminus \{1\} \\ W_{i,j}^{1^*} = W_{i,j}^1 + sgn([\mathbf{x}]_j)\, \epsilon_1,\ \forall i, j \end{cases}$$

*Proof*: See Appendix A.3.2.

Here, we provide some intuition on deriving the upper bound of the margin in the all-perturbed setting. The scheme behind this all-perturbed scenario can be viewed as an inductive layer-wise error propagation. Specifically, we can choose any perturbed layer as the commencement point of propagation, then fix any other weight matrices' values and further calculate the propagation of error from that layer using the concept in Section 3.2. In such manner, after iterating through all these weight matrices subject to weight perturbation, one could obtain the final change in output value and therefore establish the pairwise margin bound. A close inspection of the bound shows that the propagation of error causes the first term since the input layer and the rest of the terms are errors propagating since the $k$-th layer in the neural network, where $k \in [L]$.

### 3.3.2 Perturbing Multiple Layers

The all-perturbed setting is a special case of perturbing layers from an index set $I$ when $I = [L]$. We extend our analysis to the general multi-layer weight perturbation setting with $I \subseteq [L]$, which includes the single-layer ($I = \{N\}$) and all-perturbed ($I = [L]$) settings as special cases.

**Theorem 3 (multiple-layer perturbation)** *Let* $f_{\mathbf{W}}(x) = W^L(...\rho(W^1 x)...)$ *denote an L-Layer neural network. Given an index set* $I \subseteq [L]$, *we define the perturbed neural network as*
$$f_{\widehat{\mathbf{W}}}(x) = \tilde{W}^L(...\tilde{W}^N...\rho(\tilde{W}^1 x)...) \text{ with } \begin{cases} \tilde{W}^k = W^k,\ \forall k \in [L] \setminus I \\ \tilde{W}^k = \hat{W}^k,\ \hat{W}^k \in \mathbb{B}_{\mathbf{W}^k}^\infty(\epsilon_k), \forall k \in I \end{cases}$$

*Then, for any pairwise margin between* $f_{\widehat{\mathbf{W}}}^{ij}(x)$ *and* $f_{\mathbf{W}}^{ij}(x)$,

$$f_{\widehat{\mathbf{W}}}^{ij}(x) \le f_{\mathbf{W}}^{ij}(x) + \underbrace{\sum_{k \in I \setminus \{L\}} \Delta(\epsilon_k; z^{k-1^*}; f)}_{\textit{Perturbed Layer Error}} + \underbrace{\mathbb{1}(L \in I) 2\epsilon_L \left\| z^{L-1^*} \right\|_1}_{\textit{Final Layer Error}}$$

$$:= f_{\boldsymbol{W}}^{ij}(\boldsymbol{x}) + \underbrace{\eta_{\boldsymbol{W}}^{ij}(\boldsymbol{x}|I)}_{\textit{Error of Weight Perturbation}}$$

*where* $\boldsymbol{z}^{k^*} = \rho(\boldsymbol{W}^{k^*}...\rho(\boldsymbol{W}^{1^*}\boldsymbol{x})$ *with* $\boldsymbol{W}^{k^*}$ *defined as*
$$\begin{cases} W_{i,j}^{k^*} = \begin{cases} W_{i,j}^k + \epsilon_k, \ \forall i,j \ \forall k \in [L] \cap I \setminus \{1\} \\ W_{i,j}^k, \ \forall i,j \ \forall k \in [L] \setminus (I \cup \{1\}) \end{cases} \\ W_{i,j}^{1^*} = \begin{cases} W_{i,j}^1 + sgn([\boldsymbol{x}]_j)\epsilon_1, \ \forall i,j \ \ if \ 1 \in I \\ W_{i,j}^1, \ \forall i,j \quad otherwise \end{cases} \end{cases}$$

*and* $\boldsymbol{z}^{0^*} = \boldsymbol{x}$.

*Proof*: See Appendix A.3.3.

### 3.4 Surrogate Loss and Generalization Bound

#### 3.4.1 Construction of Robust Surrogate Loss on Pairwise Margin

We aim to construct a surrogate loss function based on a standard loss function and study its behavior against weight perturbations. Here we study the single-layer perturbation case and elucidate the multi-layer perturbation case in Appendix B. Specifically, given a perturbation budget $\epsilon$ for the $k$-th layer and the original loss function $\ell(f_{\mathbf{W}}(\mathbf{x}), y)$, robust training aims to minimize the following objective function: $\tilde{\ell}(f_{\boldsymbol{W}}(\mathbf{x}), y) = \max_{\hat{\mathbf{W}}^m \in \mathbb{B}_{\hat{\mathbf{W}}^m}^\infty(\epsilon)} \ell(f_{\widehat{\boldsymbol{W}}}(\mathbf{x}), y)$, which we call it as the robustness (worst-case) loss. Even for a single data point $(\mathbf{x}, y)$, it is hard to assess the exact robustness loss since it requires the maximization of a non-concave function over a norm ball. To make the problem of robust training against weight perturbations more computationally tractable, we aim to design a surrogate loss as an upper bound on the worst-case loss.

We focus on constructing a surrogate loss by means of pairwise margin bounds in Section 3.3. We first define two popular loss functions in the classification problem, ramp loss and cross entropy, and derive their surrogate versions. Define the margin function $M(f_{\boldsymbol{W}}(\mathbf{x}), y)$ as

$$M(f_{\boldsymbol{W}}(\mathbf{x}), y) = \min_{y' \neq y}[f_{\boldsymbol{W}}(\mathbf{x})]_y - [f_{\boldsymbol{W}}(\mathbf{x})]_{y'} = [f_{\boldsymbol{W}}(\mathbf{x})]_y - \max_{y' \neq y}[f_{\boldsymbol{W}}(\mathbf{x})]_{y'} \tag{1}$$

The ramp loss for a given data point $(\mathbf{x}, y)$ and neural network $f_{\boldsymbol{W}}(\cdot)$ is written as $\ell_{\mathrm{ramp}}(f_{\boldsymbol{W}}(\mathbf{x}), y) = \phi_\gamma(M(f_{\boldsymbol{W}}(\mathbf{x}), y))$, where the function $\phi_\gamma : \mathbb{R} \mapsto [0, 1]$ is defined as $\phi_\gamma(t) = 1$ if $t \leq 0$, $\phi_\gamma(t) = 0$ if $t \geq \gamma$, and $\phi_\gamma(t) = 1 - \frac{t}{\gamma}$ if $t \in [0, \gamma]$. Since the ramp loss is a piece-wise linear function, its surrogate loss can be directly obtained with the pairwise margin bound in Section 3.3. The cross entropy is written as $CE(\tilde{f}_{\boldsymbol{W}}(\mathbf{x}), y) = -\ln([\tilde{f}_{\boldsymbol{W}}(\mathbf{x})]_y)$, where $\tilde{f}_{\boldsymbol{W}}(\mathbf{x})$ represents a neural network with its output passing through a softmax layer. That is, $[\tilde{f}_{\boldsymbol{W}}(\mathbf{x})]_i = \frac{\exp^{[f_{\boldsymbol{W}}(\mathbf{x})]_i}}{\sum_{k \in [K]} \exp^{[f_{\boldsymbol{W}}(\mathbf{x})]_k}}$. For ease of demonstration, we will be using ramp loss and its pairwise margin under single-layer perturbation in the following lemma. The surrogate loss analysis for cross entropy and robust surrogate loss for multiple-layer perturbation is given in Appendix B.

**Lemma 2 (robust surrogate ramp loss)** *Let $N \in [L]$ denote the perturbed layer index and let*

$$\Psi(f_{\boldsymbol{W}}(\boldsymbol{x})) = 2 \max_{k \in [K]} \epsilon_N \left\| W_{k,:}^L \right\|_1 \Pi_{m=1}^{N-1} \|\boldsymbol{W}^m\|_{1,\infty} \Pi_{k=1}^{L-N-1} \left\| (\boldsymbol{W}^{L-k})^T \right\|_{1,\infty} \|\boldsymbol{x}\|_1$$

*be the worst case error and*

$$\hat{\ell}(f_{\boldsymbol{W}}(\boldsymbol{x}), y) := \phi_\gamma \Big\{ \underbrace{M(f_{\boldsymbol{W}}(\boldsymbol{x}), y)}_{\textit{margin}} - \underbrace{\Psi(f_{\boldsymbol{W}}(\boldsymbol{x}))}_{\textit{worst-case error}} \Big\}$$

*Then we have upper and lower bounds of $\hat{\ell}$ in terms of 0-1 losses expressed as*

$$\max_{\hat{W}^N \in \mathbb{B}_{W^N}^\infty(\epsilon)} \mathbb{1}\{y \neq \arg \max_{y' \in [K]}[f_{\hat{\boldsymbol{W}}}(\boldsymbol{x})]_{y'}\} \leq \hat{\ell}(f_{\boldsymbol{W}}(\boldsymbol{x}), y) \leq \mathbb{1}\{M(f_{\boldsymbol{W}}(\boldsymbol{x}), y) - \Psi(f_{\boldsymbol{W}}(\boldsymbol{x})) \leq \gamma\}.$$

*Proof*: Please see Appendix B.1

One could observe in the formula that the margin function $M(f_{\boldsymbol{W}}(\mathbf{x}), y)$ serves as an accuracy objective similar to the standard training process while the latter term could be conceived as the worst-case error caused by weight perturbation that should be suppressed. Therefore, by training under such an objective, we can simulate the scenario of robust training. Another intuition on the surrogate loss function is that the surrogate loss also implies the difficulty of training robust and generalizable models against large weight perturbations. Since error caused by perturbations would be surging rapidly through layers, only small perturbations can be applied in training and practice, permitting the worst-case error term to be smaller than the margin term. However, one follow-up question that naturally arises is whether or not the generalization gap will be widened when training with the robust surrogate loss. The following section investigates the generalization property while conducting robust training and provides some theoretical insights to training toward a generalizable and robust model under weight perturbation.

### 3.4.2 Generalization Gap for Robust Surrogate Loss

We consider the robust surrogate loss established in Lemma 2 and study its generalization bound via Rademacher complexity in Theorem 4, where $\mathcal{S} = \{(\mathbf{x}_i, y_i)\}_{i=1}^n$ denotes the set of i.i.d training samples, $\mathbf{X} := [\mathbf{x}_1, \mathbf{x}_2, \ldots, \mathbf{x}_n] \in \mathbb{R}^{d \times n}$ denotes the matrix composed of training data samples, and $d_{\max} = \max\{d, d_1, \ldots, d_L\}$ denotes the maximum dimension among all weight matrices. Firstly, we define the adversarial population risk and empirical risk as

$$R(f) = \mathbb{P}_{(\mathbf{x}, y) \sim D}\left\{\exists \hat{\mathbf{W}}^N \in \mathbb{B}_{\mathbf{W}^N}^{\infty}(\epsilon) \; s.t. \; y \neq \arg\max_{y' \in [K]} [f_{\hat{\mathbf{W}}}(\mathbf{x})]_{y'}\right\}$$

$$R_n(f) = \frac{1}{n}\sum_{i=1}^n \mathbb{1}\left([f_{\boldsymbol{W}}(\mathbf{x}_i)]_{y_i} \leq \gamma + \max_{y' \neq y_i}[f_{\boldsymbol{W}}(\mathbf{x}_i)]_{y'} + \Psi(f_{\boldsymbol{W}}(\mathbf{x}))\right)$$

Secondly, we denote the following empirical Rademacher complexity of both margin function and worst-case error as

$$\mathcal{R}_{\mathcal{S}}(M_{\mathcal{F}}) = \frac{4}{n^{3/2}} + \frac{60 \log(n) \log(2d_{max})}{n} \|\mathbf{X}\|_F \left(\Pi_{h=1}^L s_h\right)\left(\sum_{j=1}^L \left(\frac{b_j}{s_j}\right)^{2/3}\right)^{3/2}$$

$$\mathcal{R}_{\mathcal{S}}(\Psi_{\mathcal{F}}) = \frac{2\epsilon_N \sup_{f \in \mathcal{F}} \Pi_{m=1}^{N-1}\|\mathbf{W}^m\|_{1,\infty} \Pi_{k=0}^{L-N-1}\|(\mathbf{W}^{L-k})^T\|_{1,\infty}}{n} \|\mathbf{X}\|_{1,2}$$

**Theorem 4 (generalization gap for robust surrogate loss)** *With Lemma 2, consider the neural network hypothesis class* $\hat{\mathcal{F}} = \{f_{\widehat{\boldsymbol{W}}}(\boldsymbol{x})|\boldsymbol{W} = (\boldsymbol{W}^1, \ldots\hat{\boldsymbol{W}}^N, \ldots, \boldsymbol{W}^L), \hat{\boldsymbol{W}}^N \in \mathbb{B}_{\boldsymbol{W}^N}^{\infty}(\epsilon_N)\|\boldsymbol{W}^{\mathrm{h}}\|_{\sigma} \leq \mathrm{s_h}, \|(\boldsymbol{W}^{\mathrm{h}})^{\mathrm{T}}\|_{2,1} \leq \mathrm{b_h}, \mathrm{h} \in [\mathrm{L}]\}.$ *For any* $\gamma > 0$*, with probability at least* $1 - \delta$*, we have for all* $f_{\boldsymbol{W}}(\cdot) \in \mathcal{F}$

$$R(f) \leq R_n(f) + \frac{1}{\gamma}(\underbrace{\mathcal{R}_{\mathcal{S}}(M_{\mathcal{F}})}_{\text{standard generalization gap}} + \underbrace{\mathcal{R}_{\mathcal{S}}(\Psi_{\mathcal{F}})}_{\text{complexity term of robust training}}) + 3\sqrt{\frac{\log\frac{2}{\delta}}{2n}}$$

*Proof*: Please see Appendix C.1

As highlighted in the bracket term of Theorem 4, if the product of multiple weight norm bounds in the additional complexity term $\mathcal{R}_{\mathcal{S}}(\Psi_{\mathcal{F}})$ caused by robust training is not well confined, the model can suffer from a notable generalization gap. Consequently, our analysis suggests a solution to reduce the generalization gap by imposing norm penalty functions on all weight matrices for training generalizable neural networks subject to weight perturbations.

### 3.5 Theory-driven Loss toward Robustness and Generalization

With our theoretical insights, we now propose a robust and generalizable loss function. Standard neural network classifier training uses a classification loss $\ell_{cls}(f_{\boldsymbol{W}}(\mathbf{x}), y)$ that aims to widen the pairwise margin so as to raise accuracy, but won't necessarily be able to curb the error in output once weight perturbation is imposed. To address this issue, we propose to train under a mixed and regularized objective given a data sample $(\mathbf{x}, y)$, which would, in turn, balance the tradeoff between

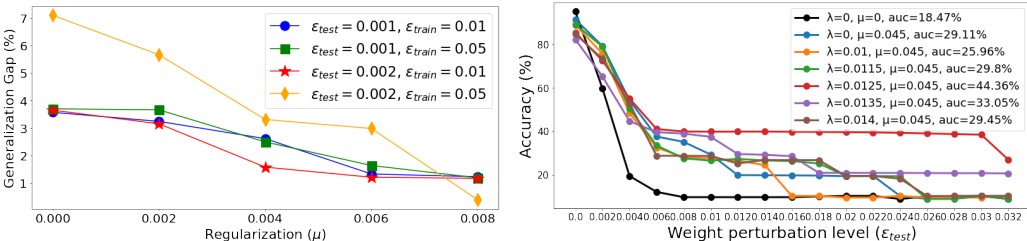

| (a) Empirical generalization gap | (b) Robust accuracy under adversarial perturbation |

Figure 1: (a) Empirical generalization gaps when varying the matrix norm regularization coefficient $\mu$ in (2) under 200-step weight PGD attack with perturbation level $\epsilon_{\text{test}}$. Consistent with the theoretical results, the gap reduces as $\mu$ increases for every $\epsilon_{\text{train}}$ value used for training. (b) Comparison of test accuracy of neural networks trained with different $\lambda$ and $\mu$ under weight PGD attack (200 steps) with perturbation level $\epsilon_{\text{test}}$. AUC refers to the area under curve score. Joint training with the two theory-driven terms in (2) indeed yields more generalizable and robust neural networks against weight perturbations.

standard accuracy, robustness, and generalization. The designed loss takes the form:

$$\ell^*(f_{\boldsymbol{W}}(\mathbf{x}), y) = \underbrace{\ell_{cls}(f_{\boldsymbol{W}}(\mathbf{x}), y)}_{\text{standard loss}} + \lambda \cdot \underbrace{\max_{y' \neq y}\{\eta_{\boldsymbol{W}}^{y'y}(\mathbf{x}|I)\}}_{\text{robustness loss from Thm. 3}} + \mu \cdot \underbrace{\sum_{m=1}^{L}\left(\left\|(\boldsymbol{W}^m)^T\right\|_{1,\infty} + \|\boldsymbol{W}^m\|_{1,\infty}\right)}_{\text{generalization gap regularization from Thm. 4}}$$

(2)

The first term in (2) originates from standard classification problem for task-specific accuracy, while the second term results from the maximum error on pairwise margin (the term $\eta_{\boldsymbol{W}}^{y'y}(x|I)$ defined in Theorem 3) induced by weight perturbation. Finally, we contribute the last term to the theoretical findings in Theorem 4, where imposing norm constraints on weight matrices could benefit generalization and prevent the generalization gap from widening. Based on our analysis, we also note that the standard technique of $L_1$ penalty function on weight applies as well.

## 4 Numerical Validation

We validate our theoretical results and the designed loss function in (2) through two sets of experiments: *empirical generalization gap with matrix norm regularization* and *robust accuracy against adversarial weight perturbations*.

**Experiment setup** We used the MNIST dataset comprised of gray-scale images of hand-written digits with ten categories. We trained neural network models as in (Section 3.1) with four dense layers (number of neurons are 128-64-32-10) and the ReLU activation function without the bias term. We used the loss function $\ell^*$ in (2) with all-layer perturbation bound (i.e., $I = [L]$), identical weight perturbation radius $\epsilon$ (or $\epsilon_{\text{train}}$), cross entropy as the standard classification loss $\ell_{cls}$, and a batch size of 32 with 20 epochs. Stochastic gradient descent with momentum is used for training, with the learning rate set to be 0.01. For the generalization experiment, we follow the same setting as in [29], which uses 1000 data samples to train the neural network. In all settings we assign identical weight perturbation budget $\epsilon$ for every layer. The comparison of weight distribution of each layer using standard and our proposed robust training is given in Appendix F. All experiments were conducted using an Intel Xeon E5-2620v4 CPU, 125 GB RAM, and an NVIDIA TITAN Xp GPU with 12 GB RAM. For reproducibility, our codes are given in the supplementary material.

**Weight PGD attack** To evaluate the robustness against weight perturbations, we modified the projected gradient descent (PGD) attack originally designed for input perturbation [12], which we call as weight PGD attack. Starting from a trained neural network weight $\boldsymbol{W}$, the perturbed weight $\widetilde{\boldsymbol{W}}$ is crafted by iterative gradient ascent using the signed gradient of the standard loss denoted as $\text{sgn}(\nabla_{\boldsymbol{W}}\ell_{cls}(f_{\widetilde{\boldsymbol{W}}}(\mathbf{x}), y))$, followed by an element-wise $\epsilon$ (or $\epsilon_{\text{test}}$) clipping centered at $\boldsymbol{W}$. The attack iteration with the step size $\alpha$ is expressed as

$$\widetilde{\boldsymbol{W}}^{(0)} = \boldsymbol{W}, \quad \widetilde{\boldsymbol{W}}^{(t+1)} = \text{Clip}_{\boldsymbol{W}, \epsilon}\left\{\widetilde{\boldsymbol{W}}^{(t)} + \alpha \, \text{sgn}(\nabla_{\boldsymbol{W}}\ell_{cls}(f_{\widetilde{\boldsymbol{W}}^{(t)}}(\mathbf{x}), y))\right\}$$

(3)

**Empirical generalization gap** Figure 1 (a) shows the empirical generalization gap (training accuracy - test accuracy) with respect to the matrix norm regularization coefficient $\mu$ under 200-step weight PGD attack with the perturbation level $\epsilon_{\text{test}}$ defined in (2) and we supplement test accuracies in Appendix D.2. As indicated in Theorem 4, increasing $\mu$ effectively suppresses the Rademacher complexity and thus reduces the generalization gaps, which are consistently observed on neural networks trained with different weight perturbation level $\epsilon_{\text{train}}$. We also conduct different settings of generalization gap analysis with $\epsilon_{\text{test}} = 0$ and different $\epsilon_{\text{train}}$ values in Appendix D.1. To demonstrate the necessity of our regularization loss function, in Appendix D.3, we run the same experiments of Figure 1 (a) but replace our regularization loss with $L_1$ and $L_2$ weight decay, respectively. It shows that only our loss function can aid reducing generalization gap and improving robustness.

**Robust accuracy against adversarial weight perturbation** We trained neural networks with different combinations of the coefficients $\lambda$ and $\mu$ in (2) using $\epsilon_{\text{train}} = 0.01$. Figure 1 (b) shows the test accuracy under different weight perturbation level $\epsilon_{\text{test}}$ (i.e., the robust accuracy) with 200 attack steps. The standard model ($\lambda = \mu = 0$) is fragile to weight PGD attack. On the other hand, neural networks trained only with the robustness loss ($\lambda > 0$ and $\mu = 0$) or the generalization gap regularization ($\lambda = 0$ and $\mu > 0$) can improve the robust accuracy due to improved generalization and classification margin. Moreover, joint training using the proposed loss with proper coefficients can further boost model performance (e.g., $(\lambda, \mu) = (0.0125, 0.045)$), as seen by the significantly improved area under curve (AUC) score of robust accuracy over all tested $\epsilon_{\text{test}}$ values. The AUC of the best model is about $2\times$ larger than that of the standard model. Similar results can be concluded for the attack with 100 steps (see Appendix D.5). Appendix D.4 shows additional experiments on the extension to convolutional neural networks trained on CIFAR-10. Similar conclusion holds. In Appendix D.6, we conduct additional experiments on the coefficients $\lambda$ and $\mu$ and discuss their tradeoffs. In summary, the results suggest the effectiveness of our theory-driven loss function for improving generalization against weight perturbation.

**Run-time analysis and weight quantization** The run-time analysis in Appendix D.8 shows the training costs for robust and standard models are comparable. In Appendix D.9, we also show that neural networks trained with our proposed loss are more robust to weight quantization. Under 2-bit quantization, the decrease in accuracy of the robust model is 3x less than the standard model.

**Non-vacuous generalization bound under weight perturbation** In the standard generalization setting, [14] showed that when increasing the size of the training data, the error bound in [4] grows rapidly, loosing the ability to describe generalization gap and thus becomes vacuous. In Appendix 3.5, we show that the error bounds of the models trained using our proposed loss function (2) in Section 3.5 would not grow in a polynomial rate and instead shows a decreasing trend, concluding the non-vacuity of the associated generalization bounds in our robust generalization setting.

**Comparison to naive adversarial weight training (AWT)** We also present comparisons of test accuracy between models trained with AWT, our loss, and the standard model under weight PGD attack with different perturbation level $\epsilon_{\text{test}}$ in Appendix G. It shows that directly applying weight PGD attack into adversarial training is not effective in training robust models.

## 5 Conclusion

In this paper, we developed a formal analysis of the robustness associated with the pairwise class margin for neural networks against weight perturbations. We also characterized its generalization gap through Rademacher complexity. A theory-driven loss function for robust generalization was proposed, and the empirical results showed significantly improved performance in generalization and robustness. Our analysis offers theoretical insights and informs the principles of training loss design toward generalizable and robust neural networks subject to weight perturbations.

## Acknowledgments

Part of this work was done during Chia-Yi Hsu's visit at IBM Research. Yu-Lin Tsai, Chia-Yi Hsu and Chia-Mu Yu were supported by MOST 110-2636-E-009-018 and we also thank to National Center for High-performance Computing (NCHC) of National Applied Research Laboratories (NARLabs) in Taiwan for providing computational and storage resources.

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
