# Appendix

## A  Margin Bound

### A.1  Toy Example

Let $f_{\mathbf{W}}(\mathbf{x}) = \mathbf{W}^3 \rho(\mathbf{W}^2 \rho(\hat{\mathbf{W}}^1 \mathbf{x}))$ denote a neural network with $\mathbf{W}^i$ being the weight matrix of the $i$-th layer and assume that one could only perturb any element in the first weight matrix within an $\ell_\infty$ norm ball of radius $\epsilon$, i.e., $\hat{\mathbf{W}}^1 \in \mathbb{B}_{\hat{\mathbf{W}}^1}^\infty(\epsilon)$. We also define an error vector as $\mathbf{e}_i$, which stands for the entry-wise error after propagating through the $i$-th layer. Since no perturbations happened prior to the first layer, we would directly take input vector $\mathbf{x}$ and derive an upper bound on the entry-wise error $\mathbf{e}_1$. While every element in the first weight matrix is allowed to change its magnitude by at most $\epsilon$, the maximum error for any entry by matrix-vector multiplication becomes

$$[\mathbf{e}_1]_i := |\hat{W}_{i,:}^1 \mathbf{x} - W_{i,:}\mathbf{x}| \leq \sum_j |\hat{W}_{i,j}^1 - W_{i,j}^1||[\mathbf{x}]_j| \leq \sum_j \epsilon|[\mathbf{x}]_j| = \epsilon \|\mathbf{x}\|_1 \qquad (4)$$

Since the following layer weight is not subject to perturbation, we simply take the magnitude of each element in the subsequent weight matrix to calculate the next error vector. In this case, we have the next layer's error $\mathbf{e}_2$ with $\mathbf{W}^2$ as $[\mathbf{e}_2]_i = \sum_j |W_{i,j}^2|[\mathbf{e}_1]_j = \epsilon \|\mathbf{x}\|_1 \sum_j |W_{i,j}^2|$. Eventually, with error propagation over layers, we arrive at the final layer and are able to assess the maximum change of any entry in output value. By recalling the pairwise class margin $f_{\mathbf{W}}^{ij}(\mathbf{x})$, we would like to inspect the relative change in error between any two classes. Specifically, we derive an upper bound on the pairwise margin between any two classes $\alpha$ and $\beta$. In the above example, the difference in entry-wise maximum error can be deduced in the following manner:

$$[\mathbf{e}_3]_\alpha - [\mathbf{e}_3]_\beta = \sum_k (|W_{\alpha,k}^3| - |W_{\beta,k}^3|)[\mathbf{e}_2]_k \overset{(i)}{\leq} \sum_k (|W_{\alpha,k}^3 - W_{\beta,k}^3|)\epsilon \|\mathbf{x}\|_1 \sum_l |W_{k,l}^2| \qquad (5)$$

$$\overset{(ii)}{\leq} \epsilon \|\mathbf{x}\|_1 \, max_k \|W_{k,:}^2\|_1 \sum_k (|W_{\alpha,k}^3 - W_{\beta,k}^3|) = \epsilon \|\mathbf{x}\|_1 \left\|(\mathbf{W}^2)^T\right\|_{1,\infty} \|W_{\alpha,:}^3 - W_{\beta,:}^3\|_1, \qquad (6)$$

where inequality $(i)$ comes from triangle inequality and inequality $(ii)$ results from taking the row in $\mathbf{W}^2$ with maximum $\ell_1$ norm. It is worth noting that there exist possible scenarios for the above inequalities to hold and therefore achieving the worst-case error. Specifically, using the example in Section 3.2, as we trace down the associated inequality bound in $(i)$, we see that the first inequality can be achieved when the final weight layer possesses all positive weights and that the row associated with label $\alpha$ is greater than label $\beta$ in all individual entries. Furthermore, as long as the second weight matrix $\mathbf{W}^2$ has equal $\ell_1$ norm throughout all rows, we can then tighten the bound to give the worst-case error in $(ii)$. Nevertheless, we could see from the above example that the difference of maximum error between entries in output would be propagating at the rate of weight matrices' $(1, \infty)$ norm.

### A.2  Single-Layer Bound

We shall first prove when $N \neq L$ and follow similar reasoning to prove the case when $N = L$. Consider the difference between set of pairwise margin $f_{\hat{\mathbf{W}}}^{ij}(x) - f_{\mathbf{W}}^{ij}(x)$, we have

$$f_{\hat{\mathbf{W}}}^{ij}(\mathbf{x}) - f_{\mathbf{W}}^{ij}(\mathbf{x})$$
$$= \{W_{i,:}^L - W_{j,:}^L\}(\hat{\mathbf{z}}^{L-1} - \mathbf{z}^{L-1}) \qquad (7)$$

$$\overset{(a)}{\leq} \|W_{i,:}^L - W_{j,:}^L\|_1 \left\|\rho(\mathbf{W}^{L-1}\hat{\mathbf{z}}^{L-2}) - \rho(\mathbf{W}^{L-1}\mathbf{z}^{L-2})\right\|_\infty \qquad (8)$$

$$\overset{(b)}{\leq} \|W_{i,:}^L - W_{j,:}^L\|_1 \left\|\mathbf{W}^{L-1}(\hat{\mathbf{z}}^{L-2} - \mathbf{z}^{L-2})\right\|_\infty \qquad (9)$$

$$\overset{(c)}{\leq} \|W_{i,:}^L - W_{j,:}^L\|_1 \left\|(\mathbf{W}^{L-1})^T\right\|_{1,\infty} \left\|(\hat{\mathbf{z}}^{L-2} - \mathbf{z}^{L-2})\right\|_\infty \qquad (10)$$

$$\overset{(d)}{\leq} \left\|W_{i,:}^L - W_{j,:}^L\right\|_1 \left\|(\mathbf{W}^{L-1})^T\right\|_{1,\infty} \cdots \left\|(\mathbf{W}^{N+1})^T\right\|_{1,\infty} \left\|(\hat{\mathbf{W}}^N - \mathbf{W}^N)\mathbf{z}^{N-1}\right\|_\infty \quad (11)$$

$$\overset{(e)}{\leq} \epsilon_N \left\|W_{i,:}^L - W_{j,:}^L\right\|_1 \left\|\mathbf{z}^{N-1}\right\|_1 \Pi_{k=1}^{L-N-1} \left\|(\mathbf{W}^{L-k})^T\right\|_{1,\infty}, \quad (12)$$

where inequality (a) results from applying Hölder inequality, and inequality (b) comes from the contractive property (1-Lipschitz) of activation function $\rho(\cdot)$. Inequality (c) and (d) come from triangle inequality applied element-wise on vector $\mathbf{W}^{L-1}(\hat{\mathbf{z}}^{L-2} - \mathbf{z}^{L-2})$ combined with induction while inequality (e) simply comes from the fact that every element in matrix $\hat{\mathbf{W}}^N - \mathbf{W}^N$ has at most $\epsilon$ in magnitude.

With similar reasoning, we proof the scenario when $N = L$ as following

$$f_{\widehat{\boldsymbol{W}}}^{ij}(\mathbf{x}) - f_{\boldsymbol{W}}^{ij}(\mathbf{x})$$
$$= \{W_{i,:}^L - W_{j,:}^L\}\mathbf{z}^{L-1} - \{\hat{W}_{i,:}^L - \hat{W}_{j,:}^L\}\mathbf{z}^{L-1} \quad (13)$$

$$\overset{(i)}{\leq} 2\epsilon_L \mathbf{1}^T \mathbf{z}^{L-1} \quad (14)$$

$$= 2\epsilon_L \left\|\mathbf{z}^{L-1}\right\|_1, \quad (15)$$

where inequality $(i)$ comes from problem definition (within element-wise $\ell_\infty$ norm ball) and since the activation function $\rho(\cdot)$ is non-negative, we could transform the inner product to its $\ell_1$ norm.

### A.3 Multi-Layer Scenario

#### A.3.1 Key Prerequisite

Before going through the proof for all-perturbed bound, we shall be introducing a maximization problem and search for its solution. Recall that we have defined in Section 3.1 the notion of output vector under weight perturbation setting, we now consider maximizing its $\ell_1$ norm over a given perturbed matrix set $\widehat{\boldsymbol{W}}$ and give the following solution. Using the notation in Section 3.1, we have $\hat{\mathbf{z}}^k$ as the output vector under perturbation setting and write the optimal vector that achieve its maximum $\ell_1$ norm as $\mathbf{z}^{k^*}$. We then obtain the following solution

$$\left\|\mathbf{z}^{k^*}\right\|_1 = \max_{\widehat{\boldsymbol{W}}} \left\|\hat{\mathbf{z}}^k\right\|_1,$$
where $\mathbf{z}^{k^*} = \rho(\mathbf{W}^{k^*}...\rho(\mathbf{W}^{1^*}\mathbf{x}))$ with $\begin{cases} \mathbf{W}_{i,j}^{m^*} = W_{i,j}^m + \epsilon_m, \ \forall i,j \ \forall m \in \{2,...,L\} \\ \mathbf{W}_{i,j}^{1^*} = W_{i,j}^1 + sgn([\mathbf{x}]_j)\epsilon_1, \ \forall i,j \end{cases}$

The reasoning behind uses the non-negative property of activation function, for any element in a matrix $\mathbf{W}$, the choice to maximize it's $\ell_1$ norm matrix-vector product is to go in the direction identical to the sign of the vector's element-wise value since the activation function is applied after the first layer, we would then obtain the solution above.

#### A.3.2 All-Perturbed Bound

In the following proof for Theorem 2, we apply similar steps in Appendix A.2 and consider the difference between set of pairwise margin under natural and weight perturbation setting, recall in Theorem 2 we defined that $f_{\boldsymbol{W}}(\mathbf{x}) = \mathbf{W}^L(..\mathbf{W}^N...\rho(\mathbf{W}^1\mathbf{x})...)$ and $f_{\widehat{\boldsymbol{W}}}(\mathbf{x}) = \hat{\mathbf{W}}^L(..\hat{\mathbf{W}}^N...\rho(\hat{\mathbf{W}}^1\mathbf{x})...)$ Thus for any set of pairwise margin $f_{\widehat{\boldsymbol{W}}}^{ij}(\mathbf{x})$ and $f_{\boldsymbol{W}}^{ij}(\mathbf{x})$, we have

$$f_{\widehat{\boldsymbol{W}}}^{ij}(\mathbf{x}) - f_{\boldsymbol{W}}^{ij}(\mathbf{x})$$
$$= \{\hat{W}_{i,:}^L - \hat{W}_{j,:}^L\}\hat{\mathbf{z}}^{L-1} - \{W_{i,:}^L - W_{j,:}^L\}\mathbf{z}^{L-1} \quad (16)$$

$$\overset{(a)}{\leq} \left\|W_{i,:}^L - W_{j,:}^L\right\|_1 \left\|\rho(\hat{\mathbf{W}}^{L-1}\hat{\mathbf{z}}^{L-2}) - \rho(\mathbf{W}^{L-1}\mathbf{z}^{L-2})\right\|_\infty + 2\epsilon_L \mathbf{1}^T \hat{\mathbf{z}}^{L-1} \quad (17)$$

$$\overset{(b)}{\leq} \left\|W_{i,:}^L - W_{j,:}^L\right\|_1 \{ \left\|\mathbf{W}^{L-1}(\hat{\mathbf{z}}^{L-2} - \mathbf{z}^{L-2})\right\|_\infty + \left\|(\hat{\mathbf{W}}^{L-1} - \mathbf{W}^{L-1})\hat{\mathbf{z}}^{L-2}\right\|_\infty \} + 2\epsilon_L \left\|\hat{\mathbf{z}}^{L-1}\right\|_1 \quad (18)$$

$$\overset{(c)}{\leq} \left\| W_{i,:}^L - W_{j,:}^L \right\|_1 \left\{ \left\| (\mathbf{W}^{L-1})^T \right\|_{1,\infty} \left\| \rho(\hat{\mathbf{W}}^{L-2} \hat{\mathbf{z}}^{L-3}) - \rho(\mathbf{W}^{L-2}\mathbf{z}^{L-3}) \right\|_\infty + \epsilon_{L-1} \left\| \hat{\mathbf{z}}^{L-2} \right\|_1 \right\}$$
$$+ 2\epsilon_L \left\| \hat{\mathbf{z}}^{L-1} \right\|_1 \tag{19}$$

$$\overset{(d)}{\leq} \left\| W_{i,:}^L - W_{j,:}^L \right\|_1 \left\{ \epsilon_1 \left\| \mathbf{x} \right\|_1 \Pi_{l=1}^{L-2} \left\| (\mathbf{W}^{L-l})^T \right\|_{1,\infty} + \sum_{j=1}^{L-3} \left( \Pi_{k=j+2}^{L-1} \left\| (\mathbf{W}^k)^T \right\|_{1,\infty} \right) \epsilon_{j+1} \left\| \hat{\mathbf{z}}^j \right\|_1 \right.$$
$$\left. + \epsilon_{L-1} \left\| \hat{\mathbf{z}}^{L-2} \right\|_1 \right\} + 2\epsilon_L \left\| \hat{\mathbf{z}}^{L-1} \right\|_1 \tag{20}$$

$$\overset{(e)}{\leq} \left\| W_{i,:}^L - W_{j,:}^L \right\|_1 \left\{ \epsilon_1 \left\| \mathbf{x} \right\|_1 \Pi_{l=1}^{L-2} \left\| (\mathbf{W}^{L-l})^T \right\|_{1,\infty} + \sum_{j=1}^{L-3} \left( \Pi_{k=j+2}^{L-1} \left\| (\mathbf{W}^k)^T \right\|_{1,\infty} \right) \epsilon_{j+1} \left\| \mathbf{z}^{j^*} \right\|_1 \right.$$
$$\left. + \epsilon_{L-1} \left\| \mathbf{z}^{L-2^*} \right\|_1 \right\} + 2\epsilon_L \left\| \mathbf{z}^{L-1^*} \right\|_1 \tag{21}$$

In the above proof, inequality (a) comes from the problem definition (perturbation of final layer within $\epsilon_L$) and inequality (b) results from the contractive property of $\rho(\cdot)$ (1-Lipschitz) combined with the use of triangle inequality. Inequality (c) was achieved through triangle inequality applied on elements of $\mathbf{W}^{L-1}(\hat{\mathbf{z}}^{L-2} - \mathbf{z}^{L-2})$ and using the fact that $\hat{\mathbf{W}}^{L-1} - \mathbf{W}^{L-1}$ has every element less than or equal to $\epsilon_{L-1}$ in magnitude. By the process of induction and maximizing the $\ell_1$ norm of perturbed output under weight perturbation $\hat{\mathbf{z}}^k$, we arrive at inequality (d) and (e).

### A.3.3 Multi-Layer Bound

We now proceed to utilize similar reasoning to establish the multi-layer bound when weight perturbation is imposed according to an index set $I$

$$f_{\tilde{\mathbf{W}}}^{ij}(\mathbf{x}) - f_{\mathbf{W}}^{ij}(\mathbf{x})$$
$$= \{\tilde{W}_{i,:}^L - \tilde{W}_{j,:}^L\}\hat{\mathbf{z}}^{L-1} - \{W_{i,:}^L - W_{j,:}^L\}\mathbf{z}^{L-1} \tag{22}$$
$$\leq \left\| W_{i,:}^L - W_{j,:}^L \right\|_1 \left\| \rho(\tilde{\mathbf{W}}^{L-1} \hat{\mathbf{z}}^{L-2}) - \rho(\mathbf{W}^{L-1}\mathbf{z}^{L-2}) \right\|_\infty + \mathbb{1}(L \in I)2\epsilon_L \mathbf{1}^T \hat{\mathbf{z}}^{L-1} \tag{23}$$
$$\leq \left\| W_{i,:}^L - W_{j,:}^L \right\|_1 \left\{ \left\| \mathbf{W}^{L-1}(\hat{\mathbf{z}}^{L-2} - \mathbf{z}^{L-2}) \right\|_\infty + \left\| (\tilde{\mathbf{W}}^{L-1} - \mathbf{W}^{L-1})\hat{\mathbf{z}}^{L-2} \right\|_\infty \right\}$$
$$+ \mathbb{1}(L \in I)2\epsilon_L \left\| \hat{\mathbf{z}}^{L-1} \right\|_1 \tag{24}$$
$$\leq \left\| W_{i,:}^L - W_{j,:}^L \right\|_1 \left\{ \left\| (\mathbf{W}^{L-1})^T \right\|_{1,\infty} \left\| \rho(\tilde{\mathbf{W}}^{L-2} \hat{\mathbf{z}}^{L-3}) - \rho(\mathbf{W}^{L-2}\mathbf{z}^{L-3}) \right\|_\infty \right.$$
$$\left. + \mathbb{1}(L-1 \in I)\epsilon_{L-1} \left\| \hat{\mathbf{z}}^{L-2} \right\|_1 \right\} + \mathbb{1}(L \in I)2\epsilon_L \left\| \hat{\mathbf{z}}^{L-1} \right\|_1 \tag{25}$$
$$\leq \left\| W_{i,:}^L - W_{j,:}^L \right\|_1 \left\{ \mathbb{1}(1 \in I)\epsilon_1 \left\| \mathbf{x} \right\|_1 \Pi_{l=1}^{L-2} \left\| (\mathbf{W}^{L-l})^T \right\|_{1,\infty} + \mathbb{1}(L-1 \in I)\epsilon_{L-1} \left\| \hat{\mathbf{z}}^{L-2} \right\|_1 \right.$$
$$\left. + \sum_{j=1}^{L-3} \mathbb{1}(j+1 \in I)\left( \Pi_{k=j+2}^{L-1} \left\| (\mathbf{W}^k)^T \right\|_{1,\infty} \right) \epsilon_{j+1} \left\| \hat{\mathbf{z}}^j \right\|_1 \right\} + \mathbb{1}(L \in I)2\epsilon_L \left\| \hat{\mathbf{z}}^{L-1} \right\|_1 \tag{26}$$
$$\leq \left\| W_{i,:}^L - W_{j,:}^L \right\|_1 \left\{ \sum_{\ell \in I \setminus \{L, L-1\}} \left( \Pi_{k=\ell+1}^{L-1} \left\| (\mathbf{W}^k)^T \right\|_{1,\infty} \right) \epsilon_\ell \left\| \mathbf{z}^{\ell-1^*} \right\|_1 \right.$$
$$\left. + \mathbb{1}(L-1 \in I)\epsilon_{L-1} \left\| \mathbf{z}^{L-2^*} \right\|_1 \right\} + \mathbb{1}(L \in I)2\epsilon_L \left\| \mathbf{z}^{L-1^*} \right\|_1 \tag{27}$$

The proof for multi-layer bound follows same reasoning from the all-perturbed setting except indicator function was added to check whether a certain layer $m$ is in the index set $I$ and at last we rewrite the expression using the members of set $I$.

## A.4 Convolutional Neural Network

We now offer a similar proof for convolutional neural networks. Consider a network structure where there are $N$ convolution layers followed by $M$ dense layer for a $K$-class classification problem with activation function $\rho(\cdot)$ applied after each layer. We note that each convolution operation can be described as matrix multiplication of a doubly block Toeplitz matrix. With such formulation, we present the theorem as follows

**Theorem 5 (all-layer perturbation)** *Let* $f_{\boldsymbol{W}}(\boldsymbol{x}_0) = \boldsymbol{W}^{M+N}(...\rho(\boldsymbol{W}^{N+1}\boldsymbol{z})...)$ *and* $\boldsymbol{z} = \rho(\boldsymbol{T}^N...\rho(\boldsymbol{T}^1\boldsymbol{x}_0))$ *where* $\boldsymbol{T}^i$ *stands for the Toeplitz matrix of convolution operation in the $i$-th layer, denote an composite $M+N$-Layer neural network. We further define the perturbed neural network as* $f_{\widehat{\boldsymbol{W}}}(\boldsymbol{x}_0) = \widetilde{\boldsymbol{W}}^{M+N}(...\rho(\widetilde{\boldsymbol{W}}^{N+1}\tilde{z})...)$ *and* $\tilde{z} = \rho(\widetilde{\boldsymbol{T}}^N...\rho(\widetilde{\boldsymbol{T}}^1\boldsymbol{x}_0))$ *Then, if* $\boldsymbol{x}_0 \in [0,1]^d$, *for any pairwise margin between* $f_{\widehat{\boldsymbol{W}}}^{ij}(\boldsymbol{x}_0)$ *and* $f_{\boldsymbol{W}}^{ij}(\boldsymbol{x}_0)$,

$$f_{\widehat{\boldsymbol{W}}}^{ij}(\boldsymbol{x}_0) \leq f_{\boldsymbol{W}}^{ij}(\boldsymbol{x}_0) + \underbrace{\sum_{k=1}^{M+N-1} \Delta(\epsilon_k; \boldsymbol{z}^{k-1^*}; f)}_{\text{Dense Layer Error}} + \underbrace{2\epsilon_{M+N} \left\| \boldsymbol{z}^{M+N-1^*} \right\|_1}_{\text{Final Layer Error}}$$

*where* $\boldsymbol{z}^{k^*} = \rho(\boldsymbol{W}^{k^*}...\rho(\boldsymbol{T}^{1^*}\boldsymbol{x}))$ *with* $\boldsymbol{T}^{i^*}$ *and* $\boldsymbol{W}^{j^*}$ *defined as* $\begin{cases} \boldsymbol{T}_{r,s}^{i^*} = \boldsymbol{T}_{r,s}^i + \epsilon_i \\ \boldsymbol{W}_{r,s}^{j^*} = \boldsymbol{W}_{r,s}^j + \epsilon_j \end{cases}$

*and* $\boldsymbol{z}^{0^*} = \boldsymbol{x}_0.$

*Proof:*

$$f_{\widehat{\boldsymbol{W}}}^{ij}(\mathbf{x}_0) - f_{\boldsymbol{W}}^{ij}(\mathbf{x}_0)$$
$$= \{\hat{W}_{i,:}^{M+N} - \hat{W}_{j,:}^{M+N}\}\hat{\mathbf{z}}^{M+N-1} - \{W_{i,:}^{M+N} - W_{j,:}^{M+N}\}\mathbf{z}^{M+N-1} \tag{28}$$
$$\leq \left\| W_{i,:}^{M+N} - W_{j,:}^{M+N} \right\|_1 \left\| \rho(\hat{\mathbf{W}}^{M+N-1}\hat{\mathbf{z}}^{M+N-2}) - \rho(\mathbf{W}^{M+N-1}\mathbf{z}^{M+N-2}) \right\|_\infty + 2\epsilon_{M+N}\mathbf{1}^T\hat{\mathbf{z}}^{M+N-1} \tag{29}$$
$$\leq \left\| W_{i,:}^{M+N} - W_{j,:}^{M+N} \right\|_1 \left\{ \left\| \mathbf{W}^{M+N-1}(\hat{\mathbf{z}}^{M+N-2} - \mathbf{z}^{M+N-2}) \right\|_\infty + \left\| (\hat{\mathbf{W}}^{M+N-1} - \mathbf{W}^{M+N-1})\hat{\mathbf{z}}^{M+N-2} \right\|_\infty \right\}$$
$$+ 2\epsilon_{M+N} \left\| \hat{\mathbf{z}}^{M+N-1} \right\|_1 \tag{30}$$
$$\leq \left\| W_{i,:}^{M+N} - W_{j,:}^{M+N} \right\|_1 \left\{ \left\| (\mathbf{W}^{M+N-1})^T \right\|_{1,\infty} \left\| \rho(\hat{\mathbf{W}}^{M+N-2}\hat{\mathbf{z}}^{M+N-3}) - \rho(\mathbf{W}^{M+N-2}\mathbf{z}^{M+N-3}) \right\|_\infty \right.$$
$$\left. + \epsilon_{M+N-1} \left\| \hat{\mathbf{z}}^{M+N-2} \right\|_1 \right\} + 2\epsilon_{M+N} \left\| \hat{\mathbf{z}}^{M+N-1} \right\|_1 \tag{31}$$
$$\leq \left\| W_{i,:}^{M+N} - W_{j,:}^{M+N} \right\|_1 \left\{ \epsilon_1 \left\| \mathbf{x}_0 \right\|_1 \Pi_{s=1}^N \left\| (\mathbf{T})^T \right\|_{1,\infty} \Pi_{l=N+1}^{M+N-2} \left\| (\mathbf{W}^{L-l})^T \right\|_{1,\infty} \right.$$
$$+ \sum_{j=1}^{M+N-3} \left( \Pi_{k=j+2}^{M+N-1} \left\| (\mathbf{A}^k)^T \right\|_{1,\infty} \right) \epsilon_{j+1} \left\| \hat{\mathbf{z}}^j \right\|_1 + \epsilon_{L-1} \left\| \hat{\mathbf{z}}^{M+N-2} \right\|_1 \right\} + 2\epsilon_{M+N} \left\| \hat{\mathbf{z}}^{M+N-1} \right\|_1 \tag{32}$$
$$\leq \left\| W_{i,:}^{M+N} - W_{j,:}^{M+N} \right\|_1 \left\{ \epsilon_1 \left\| \mathbf{x}_0 \right\|_1 \Pi_{s=1}^N \left\| (\mathbf{T})^T \right\|_{1,\infty} \Pi_{l=N+1}^{M+N-2} \left\| (\mathbf{W}^{L-l})^T \right\|_{1,\infty} \right.$$
$$+ \sum_{j=1}^{M+N-3} \left( \Pi_{k=j+2}^{M+N-1} \left\| (\mathbf{A}^k)^T \right\|_{1,\infty} \right) \epsilon_{j+1} \left\| \mathbf{z}^{j^*} \right\|_1 + \epsilon_{L-1} \left\| \mathbf{z}^{M+N-2^*} \right\|_1 \right\} + 2\epsilon_{M+N} \left\| \mathbf{z}^{M+N-1^*} \right\|_1 \tag{33}$$

Notice that in the proof $\mathbf{A}^k$ is defined as $\mathbf{A}^k = \begin{cases} \mathbf{W}^k & k \geq N+1 \\ \mathbf{T}^k & k < N+1 \end{cases}$

# B  Surrogate Loss

## B.1  Case on Ramp Loss

We now provide a proof for Lemma 2. Recall the definition of ramp function in Section 3.4.1, we have that ramp loss for a given data point $(\mathbf{x}, y)$ and neural network $f_{\boldsymbol{W}}(\cdot)$ is written as $\ell_{\mathrm{ramp}}(f_{\boldsymbol{W}}(\mathbf{x}), y) = \phi_\gamma(M(f_{\boldsymbol{W}}(\mathbf{x}), y))$, where the function $\phi_\gamma : \mathbb{R} \mapsto [0, 1]$ is defined as

$$\phi_\gamma(t) = \begin{cases} 1 & if \quad t \leq 0 \\ 0 & if \quad t \geq \gamma \\ 1 - \frac{t}{\gamma} & if \quad t \in [0, \gamma] \end{cases} \tag{34}$$

Then for any $(\mathbf{x}, y)$, using ReLU as activation function, we have

$$\max_{\widehat{\boldsymbol{W}}} \mathbb{1}(y \neq \arg\max_{y'}[f_{\widehat{\boldsymbol{W}}}(\mathbf{x})]_{y'})$$

$$\overset{(a)}{\leq} \phi_\gamma(\min_{\widehat{\boldsymbol{W}}} M(f_{\widehat{\boldsymbol{W}}}(\mathbf{x}), y)) \tag{35}$$

$$\overset{(b)}{\leq} \phi_\gamma(\min_{y' \neq y} \min_{\widehat{\boldsymbol{W}}}[f_{\widehat{\boldsymbol{W}}}(\mathbf{x})]_y - [f_{\widehat{\boldsymbol{W}}}(\mathbf{x})]_{y'}) \tag{36}$$

$$\overset{(c)}{\leq} \phi_\gamma\big( \min_{y' \neq y}[f_{\boldsymbol{W}}(\mathbf{x})]_y - [f_{\boldsymbol{W}}(\mathbf{x})]_{y'} - \max_{y' \neq y} \epsilon_N \left\| W_{y',:}^L - W_{y,:}^L \right\|_1 \left\| \mathbf{z}^{N-1} \right\|_1 \Pi_{k=1}^{L-N-1} \left\| (\mathbf{W}^{L-k})^T \right\|_{1,\infty} \big) \tag{37}$$

$$\overset{(d)}{\leq} \phi_\gamma\big( \min_{y' \neq y}[f_{\boldsymbol{W}}(\mathbf{x})]_y - [f_{\boldsymbol{W}}(\mathbf{x})]_{y'} - 2 \max_{k \in [K]} \epsilon_N \left\| W_{k,:}^L \right\|_1 \left\| \mathbf{z}^{N-1} \right\|_1 \Pi_{k=1}^{L-N-1} \left\| (\mathbf{W}^{L-k})^T \right\|_{1,\infty} \big) \tag{38}$$

$$\overset{(e)}{\leq} \phi_\gamma\big( M(f_{\boldsymbol{W}}(\mathbf{x}), y) - 2 \max_{k \in [K]} \epsilon_N \left\| W_{k,:}^L \right\|_1 \left\| \mathbf{z}^{N-1} \right\|_1 \Pi_{k=1}^{L-N-1} \left\| (\mathbf{W}^{L-k})^T \right\|_{1,\infty} \big) \tag{39}$$

$$\overset{(f)}{\leq} \phi_\gamma\big( M(f_{\boldsymbol{W}}(\mathbf{x}), y) - 2 \max_{k \in [K]} \epsilon_N \left\| W_{k,:}^L \right\|_1 \left\| \mathbf{x} \right\|_1 \Pi_{m=1}^{N-1} \left\| \mathbf{W}^m \right\|_{1,\infty} \Pi_{k=1}^{L-N-1} \left\| (\mathbf{W}^{L-k})^T \right\|_{1,\infty} \big) \tag{40}$$

$$:= \hat{\ell}(f_{\boldsymbol{W}}(\mathbf{x}), y) \tag{41}$$

$$\overset{(g)}{\leq} \mathbb{1}\big( M(f_{\boldsymbol{W}}(\mathbf{x}), y) - 2 \max_{k \in [K]} \epsilon_N \left\| W_{k,:}^L \right\|_1 \left\| \mathbf{x} \right\|_1 \Pi_{m=1}^{N-1} \left\| \mathbf{W}^m \right\|_{1,\infty} \Pi_{k=1}^{L-N-1} \left\| (\mathbf{W}^{L-k})^T \right\|_{1,\infty} \leq \gamma \big), \tag{42}$$

where inequality (a) is due to the property of ramp loss while inequality (b) is by the definition of margin and inequality (c) comes from applying Theorem 1. Inequality (d) results from using triangle inequality and taking its maximum, inequality (e) is by the definition of margin and inequality (f) comes from the fact that with ReLU we have $\| \rho(\mathbf{Ax}) \|_1 \leq \| \mathbf{Ax} \|_1$. Lastly, inequality (g) is a direct consequence from property of ramp loss.

### B.1.1  Ramp Loss on Multiple Layer Bound

We now follow a similar course and prove robust ramp loss using the multi-layer bound in Theorem 3. We consider the robust loss form proposed in Section 3.4.1 and have that,

$$\max_{\widehat{\boldsymbol{W}}} \ell_{\mathrm{ramp}}(f_{\widehat{\boldsymbol{W}}}(\mathbf{x}), y)$$

$$\overset{(a)}{\leq} \phi_\gamma(\min_{\widehat{\boldsymbol{W}}} M(f_{\widehat{\boldsymbol{W}}}(\mathbf{x}), y)) \tag{43}$$

$$\overset{(b)}{\leq} \phi_\gamma\big( \min_{y' \neq y}[f_{\boldsymbol{W}}(\mathbf{x})]_y - [f_{\boldsymbol{W}}(\mathbf{x})]_{y'} - \max_{y' \neq y} \eta_{\boldsymbol{W}}^{y'y}(\mathbf{x}|I) \big) \tag{44}$$

$$\overset{(c)}{\leq} \phi_\gamma\big( M(f_{\boldsymbol{W}}(\mathbf{x}), y) - \max_{y' \neq y} \eta_{\boldsymbol{W}}^{y'y}(\mathbf{x}|I) \big) := \hat{\ell}(f_{\boldsymbol{W}}(\mathbf{x}), y) \tag{45}$$

## B.2 Cross Entropy

We further consider the case of cross entropy and prove an upper bound for it. We denote the loss function as $CE(\cdot)$, and during training, hard label was applied. Recall the definition of $\tilde{f}_{\boldsymbol{W}}(\mathbf{x})$ in Section 3.4.1, we have the difference of loss function between natural and perturbation settings as,

$$CE(\tilde{f}_{\widehat{\boldsymbol{W}}}(\mathbf{x}), y) - CE(\tilde{f}_{\boldsymbol{W}}(\mathbf{x}), y)$$

$$= -y \ln \frac{[\tilde{f}_{\widehat{\boldsymbol{W}}}(\mathbf{x})]_y}{[\tilde{f}_{\boldsymbol{W}}(\mathbf{x})]_y} \tag{46}$$

$$= \ln \frac{[\tilde{f}_{\boldsymbol{W}}(\mathbf{x})]_y}{[\tilde{f}_{\widehat{\boldsymbol{W}}}(\mathbf{x})]_y} \tag{47}$$

$$= \ln \left( e^{[f_{\boldsymbol{W}}(\mathbf{x})]_y - [f_{\widehat{\boldsymbol{W}}}(\mathbf{x})]_y} \frac{\sum_{k \in [K]} e^{[f_{\widehat{\boldsymbol{W}}}(\mathbf{x})]_k}}{\sum_{k \in [K]} e^{[f_{\boldsymbol{W}}(\mathbf{x})]_k}} \right\} \tag{48}$$

$$\overset{(a)}{\le} \ln \left( \max_{y' \neq y} e^{[f_{\boldsymbol{W}}(\mathbf{x})]_y - [f_{\widehat{\boldsymbol{W}}}(\mathbf{x})]_y} \cdot e^{[f_{\widehat{\boldsymbol{W}}}(\mathbf{x})]_{y'} - [f_{\boldsymbol{W}}(\mathbf{x})]_{y'}} \right) \tag{49}$$

$$\overset{(b)}{\le} \ln \left( e^{\max_{y' \neq y} f_{\widehat{\boldsymbol{W}}}^{y'y}(\mathbf{x}) - f_{\boldsymbol{W}}^{y'y}(\mathbf{x})} \right) \tag{50}$$

$$\overset{(c)}{=} \max_{y' \neq y} \eta_{\boldsymbol{W}}^{y'y}(\mathbf{x}|I), \tag{51}$$

where inequality (a) comes from taking the maximum in the set of all ratios $\left\{ \frac{e^{[f_{\widehat{\boldsymbol{W}}}(\mathbf{x})]_k}}{e^{[f_{\boldsymbol{W}}(\mathbf{x})]_k}} \right\}$ and inequality (b) comes from monotonicity of exponential function. Finally, the last expression (c) can be referred to Theorem 3. Thus with the above proof, we could establish the following robust surrogate loss for cross entropy, denoted as $\widehat{CE}(f_{\boldsymbol{W}}(\mathbf{x}), y)$ :

$$\widehat{CE}(f_{\boldsymbol{W}}(\mathbf{x}), y) = CE(f_{\boldsymbol{W}}(\mathbf{x}), y) + \max_{y' \neq y} \eta_{\boldsymbol{W}}^{y'y}(\mathbf{x}|I)$$

# C  Generalization Bound on Rademacher Complexity

## C.1  Proof on Single Layer Bound

To show the Rademacher complexity and generalization gap on single layer robust surrogate loss, we first introduce a result proven in [4] and another classical result in statistical learning theory and proceed to give a proof on Theorem 4. Given a set $\mathcal{S} = \{(\mathbf{x}_i, y_i)\}_{i=1}^n$ of i.i.d training samples, denote $\mathbf{X} := [\mathbf{x}_1, \mathbf{x}_2, \ldots, \mathbf{x}_n] \in \mathbb{R}^{d \times n}$ as the matrix composed of training data and let $d_{\max} = \max\{d, d_1, \ldots, d_L\}$ as the maximum dimension among all weight matrices.

**Lemma 3 ([13])** *Assume that the range of loss function $\ell(\cdot)$ is [0,1]. Then, for any $\delta \in (0, 1)$, with probability as least $1 - \delta$, we have for all $f \in \mathcal{F}$*

$$R(f) \le R_n(f) + 2\mathcal{R}(\ell_{\mathcal{F}}) + 3\sqrt{\frac{\log \frac{2}{\delta}}{2n}},$$

*where $R(f)$ and $R_n(f)$ stand for population risk and empirical risk, respectively.*

**Lemma 4 ([4])** *Consider the neural network hypothesis class,*

$$\mathcal{F} = \{f_{\boldsymbol{W}}(\boldsymbol{x}) \mid \boldsymbol{W} = (\boldsymbol{W}^1, \boldsymbol{W}^2, ..., \boldsymbol{W}^L), \left\|\boldsymbol{W}^h\right\|_\sigma \le s_h, \left\|(\boldsymbol{W}^h)^T\right\|_{2,1} \le b_h \, h \in [L]\}$$

*We have an upper bound on the Rademacher complexity,*

$$\mathcal{R}(\mathcal{F}) \le \frac{4}{n^{3/2}} + \frac{26 \log(n) \log(2d_{max})}{n} \left\|\boldsymbol{X}\right\|_F \left( \Pi_{h=1}^L s_h \right) \left( \sum_{j=1}^L \left( \frac{b_j}{s_j} \right)^{2/3} \right)^{3/2}$$

We now study the Rademacher Complexity of the function class

$$\hat{\ell}_{\mathcal{F}} = \{(\mathbf{x}, y) \mapsto \hat{\ell}(f_{\boldsymbol{W}}(\mathbf{x}), y) | f \in \mathcal{F}\},$$

where $\hat{\ell}(\cdot)$ is denoted in Lemma 2 and let $M_{\mathcal{F}} = \{(\mathbf{x}, y) \mapsto M(f_{\boldsymbol{W}}(\mathbf{x}), y) | f \in \mathcal{F}\}$. Then we could obtain,

$$\mathcal{R}(\hat{\ell}_{\mathcal{F}}) \leq \frac{1}{\gamma}\Big(\mathcal{R}(M_{\mathcal{F}}) + \frac{2\epsilon_N}{n} E_\nu[\sup_{f \in \mathcal{F}} \sum_{i=1}^n \nu_i \max_{k \in [K]} \|W_{k,:}^L\|_1 \Pi_{m=1}^{N-1} \|\mathbf{W}^m\|_1 \Pi_{k=1}^{L-N-1} \|(\mathbf{W}^{L-k})^T\|_{1,\infty} \|\mathbf{x}_i\|_1]\Big), \tag{52}$$

where the inequality was achieved by using the Ledoux-Talagrand contraction inequality and the convexity of the supreme operation. Consider the second term, we have that

$$\frac{2\epsilon_N}{n} E_\nu[\sup_{f \in \mathcal{F}} \sum_{i=1}^n \nu_i \max_{k \in [K]} \|W_{k,:}^L\|_1 \Pi_{m=1}^{N-1} \|\mathbf{W}^m\|_1 \Pi_{k=1}^{L-N-1} \|(\mathbf{W}^{L-k})^T\|_{1,\infty} \|\mathbf{x}_i\|_1] \tag{53}$$

$$\overset{(a)}{\leq} \frac{2\epsilon_N}{n}\Big(\sup_{f \in \mathcal{F}} \max_{k \in [K]} \|W_{k,:}^L\|_1 \Pi_{m=1}^{N-1} \|\mathbf{W}^m\|_1 \Pi_{k=1}^{L-N-1} \|(\mathbf{W}^{L-k})^T\|_{1,\infty}\Big) E_\nu[|\sum_{i=1}^n \nu_i \|\mathbf{x}_i\|_1|] \tag{54}$$

$$\overset{(b)}{\leq} \frac{2\epsilon_N}{n}\Big(\sup_{f \in \mathcal{F}} \Pi_{m=1}^{N-1} \|\mathbf{W}^m\|_1 \Pi_{k=0}^{L-N-1} \|(\mathbf{W}^{L-k})^T\|_{1,\infty}\Big) \|\mathbf{X}\|_{1,2}, \tag{55}$$

where inequality (a) is achieved by separating all neural network related parameters and inequality (b) is a result of applying Khintchine's inequality.

Thus, combined with Lemma 4, we have that

$$\mathcal{R}(\hat{\ell}_{\mathcal{F}}) \leq \frac{1}{\gamma}\Big(\frac{4}{n^{3/2}} + \frac{60 \log(n) \log(2d_{max})}{n} \|\mathbf{X}\|_F \left(\Pi_{h=1}^L s_h\right)\left(\sum_{j=1}^L \big(\frac{b_j}{s_j}\big)^{2/3}\right)^{3/2}$$

$$+ \frac{2\epsilon_N}{n}\big(\sup_{f \in \mathcal{F}} \Pi_{m=1}^{N-1} \|\mathbf{W}^m\|_1 \Pi_{k=0}^{L-N-1} \|(\mathbf{W}^{L-k})^T\|_{1,\infty}\big) \|\mathbf{X}\|_{1,2} \tag{56}$$

Once we have calculated an upper bound for $\mathcal{R}(\hat{\ell}_{\mathcal{F}})$, then Theorem 4 is a direct consequence of Lemma 2 and 3.

## C.2 Extension to Multiple Layer Bound

In this section, we consider the robust surrogate loss under multi-layer bound and study its Rademacher complexity, We first give the expression of the robust surrogate loss then give an result on generalization bound.

**Lemma 5** *Define the robust loss function $\hat{\ell}(f_{\boldsymbol{W}}(\boldsymbol{x}), y)$ as*

$$\hat{\ell}(f_{\boldsymbol{W}}(\boldsymbol{x}), y) = \phi_\gamma\Bigg(M(f_{\boldsymbol{W}}(\boldsymbol{x}), y)$$

$$- 2 \max_{k \in [K]} \|W_{k,:}^L\|_1 \Bigg\{ \sum_{\ell \in I \setminus \{L, L-1\}} \epsilon_\ell\big(\Pi_{i=1}^{l-1} \|\boldsymbol{W}^{i^*}\|_{1,\infty}\big)\big(\Pi_{j=\ell+1}^{L-1} \|(\boldsymbol{W}^j)^T\|_{1,\infty}\big) \|\boldsymbol{x}\|_1$$

$$+ \mathbb{1}(L-1 \in I)\epsilon_{L-1}\big(\Pi_{i=1}^{L-2} \|\boldsymbol{W}^{i^*}\|_{1,\infty}\big) \|\boldsymbol{x}\|_1 \Bigg\} - \mathbb{1}(L \in I)2\epsilon_L\big(\Pi_{i=1}^{L-1} \|\boldsymbol{W}^{i^*}\|_{1,\infty}\big) \|\boldsymbol{x}\|_1 \Bigg) \tag{57}$$

*We would have that*

$$\max_{\widehat{\boldsymbol{W}}} \mathbb{1}(y \neq \arg\max_{y'}[f_{\widehat{\boldsymbol{W}}}(\boldsymbol{x})]_{y'}) \leq \hat{\ell}(f_{\boldsymbol{W}}(\boldsymbol{x}), y)$$

$$\leq \mathbb{1}\Bigg(M(f_{\boldsymbol{W}}(\boldsymbol{x}), y) - 2 \max_{k \in [K]} \|W_{k,:}^L\|_1 \Bigg\{ \sum_{\ell \in I \setminus \{L, L-1\}} \epsilon_\ell\big(\Pi_{i=1}^{l-1} \|\boldsymbol{W}^{i^*}\|_{1,\infty}\big)\big(\Pi_{j=\ell+1}^{L-1} \|(\boldsymbol{W}^j)^T\|_{1,\infty}\big) \|\boldsymbol{x}\|_1$$

$$+ \mathbb{1}(L-1 \in I)\epsilon_{L-1}\big(\Pi_{i=1}^{L-2} \|\boldsymbol{W}^{i^*}\|_{1,\infty}\big) \|\boldsymbol{x}\|_1 \Bigg\} - \mathbb{1}(L \in I)2\epsilon_L\big(\Pi_{i=1}^{L-1} \|\boldsymbol{W}^{i^*}\|_{1,\infty}\big) \|\boldsymbol{x}\|_1 \leq \gamma\Bigg)$$

Using the above loss, we could further establish an upper bound on robust surrogate loss and provide statements on generalization bound. Given the following function class

$$\hat{\ell}_{\mathcal{F}} = \{(\mathbf{x}, y) \mapsto \hat{\ell}(f_{\mathbf{W}}(\mathbf{x}), y) | f \in \mathcal{F}\}$$

We have that,

$$\mathcal{R}(\hat{\ell}_{\mathcal{F}}) \leq \frac{1}{\gamma}\Big(\mathcal{R}(M_{\mathcal{F}}) + \frac{2}{n}E_{\nu}\Big[\sup_{f \in \mathcal{F}} \sum_{i=1}^{n} \nu_i \max_{k \in [K]} \|W_{k,:}^L\|_1 \{ \sum_{\ell \in I \setminus \{L, L-1\}} \epsilon_{\ell}(\Pi_{i=1}^{l-1}\|\mathbf{W}^{i^*}\|_{1,\infty}) \times$$

$$(\Pi_{j=\ell+1}^{L-1}\|(\mathbf{W}^j)^T\|_{1,\infty}) \|\mathbf{x}_i\|_1 + \mathbb{1}(L-1 \in I)\epsilon_{L-1}(\Pi_{i=1}^{L-2}\|\mathbf{W}^{i^*}\|_{1,\infty}) \|\mathbf{x}_i\|_1 \}\Big]$$

$$+ \frac{2}{n}E_{\nu}\Big[\sup_{f \in \mathcal{F}} \sum_{i=1}^{n} \nu_i \mathbb{1}(L \in I)\epsilon_L(\Pi_{i=1}^{L-1}\|\mathbf{W}^{i^*}\|_{1,\infty}) \|\mathbf{x}_i\|_1\Big]\Big) \tag{58}$$

which the second term can be bounded as,

$$\frac{2}{n}E_{\nu}\Big[\sup_{f \in \mathcal{F}} \sum_{i=1}^{n} \nu_i \max_{k \in [K]} \|W_{k,:}^L\|_1 \{ \sum_{\ell \in I \setminus \{L, L-1\}} \epsilon_{\ell}(\Pi_{i=1}^{l-1}\|\mathbf{W}^{i^*}\|_{1,\infty}) \times$$

$$(\Pi_{j=\ell+1}^{L-1}\|(\mathbf{W}^j)^T\|_{1,\infty}) \|\mathbf{x}_i\|_1 + \mathbb{1}(L-1 \in I)\epsilon_{L-1}(\Pi_{i=1}^{L-2}\|\mathbf{W}^{i^*}\|_{1,\infty}) \|\mathbf{x}_i\|_1 \}\Big] \tag{59}$$

$$\leq \frac{2}{n}\sup_{f \in \mathcal{F}}\max_{k \in [K]} \{ \|W_{k,:}^L\|_1 \{ \sum_{\ell \in I \setminus \{L, L-1\}} \epsilon_{\ell}(\Pi_{i=1}^{l-1}\|\mathbf{W}^{i^*}\|_{1,\infty})(\Pi_{j=\ell+1}^{L-1}\|(\mathbf{W}^j)^T\|_{1,\infty})$$

$$+ \mathbb{1}(L-1 \in I)\epsilon_{L-1}(\Pi_{i=1}^{L-2}\|\mathbf{W}^{i^*}\|_{1,\infty})\}\}E_{\nu}[|\sum_{i=1}^{n} \nu_i \|\mathbf{x}_i\|_1|] \tag{60}$$

$$\leq \frac{2}{n}\sup_{f \in \mathcal{F}}\max_{k \in [K]} \{ \|W_{k,:}^L\|_1 \{ \sum_{\ell \in I \setminus \{L, L-1\}} \epsilon_{\ell}(\Pi_{i=1}^{l-1}\|\mathbf{W}^{i^*}\|_{1,\infty})(\Pi_{j=\ell+1}^{L-1}\|(\mathbf{W}^j)^T\|_{1,\infty})$$

$$+ \mathbb{1}(L-1 \in I)\epsilon_{L-1}(\Pi_{i=1}^{L-2}\|\mathbf{W}^{i^*}\|_{1,\infty})\}\}\|\mathbf{X}\|_{1,2} \tag{61}$$

while the last term can as well be bounded as,

$$\frac{2}{n}E_{\nu}\Big[\sup_{f \in \mathcal{F}} \sum_{i=1}^{n} \nu_i \mathbb{1}(L \in I)\epsilon_L(\Pi_{i=1}^{L-1}\|\mathbf{W}^{i^*}\|_{1,\infty}) \|\mathbf{x}_i\|_1\Big] \tag{62}$$

$$\leq \frac{2\sup_{f \in \mathcal{F}}\mathbb{1}(L \in I)\epsilon_L(\Pi_{i=1}^{L-1}\|\mathbf{W}^{i^*}\|_{1,\infty})}{n}E_{\nu}[|\sum_{i=1}^{n} \nu_i \|\mathbf{x}_i\|_1|] \tag{63}$$

$$\leq \frac{2\mathbb{1}(L \in I)\epsilon_L \sup_{f \in \mathcal{F}}(\Pi_{i=1}^{L-1}\|\mathbf{W}^{i^*}\|_{1,\infty})}{n}\|\mathbf{X}\|_{1,2} \tag{64}$$

With all of the upper bounds above and Lemma 4, 5, we have the following theorem,

**Theorem 6 (generalization gap for robust surrogate loss)** *With Lemma 5, consider the neural network hypothesis class $\mathcal{F} = \{f_{\mathbf{W}}(\boldsymbol{x}) | \mathbf{W} = (\mathbf{W}^1, \mathbf{W}^2, ..., \mathbf{W}^L), \|\mathbf{W}^h\|_{\sigma} \leq s_h, \|(\mathbf{W}^h)^T\|_{2,1} \leq b_h, h \in [L]\}$. For any $\gamma > 0$, with probability at least $1 - \delta$, we have for all $f_{\mathbf{W}}(\cdot) \in \mathcal{F}$*

$$\mathbb{P}_{(\boldsymbol{x}, y) \sim D}\{\exists \widehat{\mathbf{W}} \ s.t. \ y \neq \arg\max_{y' \in [K]} [f_{\widehat{\mathbf{W}}}(\boldsymbol{x})]_{y'}\} \leq \frac{1}{n}\sum_{i=1}^{n} \mathbb{1}\Big([f_{\mathbf{W}}(\boldsymbol{x}_i)]_{y_i} \leq \gamma + \max_{y' \neq y_i}[f_{\mathbf{W}}(\boldsymbol{x}_i)]_{y'} + 2\max_{k \in [K]}\|W_{k,:}^L\|_1$$

$$\times \{ \sum_{\ell \in I \setminus \{L, L-1\}} \epsilon_{\ell}(\Pi_{i=1}^{l-1}\|\mathbf{W}^{i^*}\|_{1,\infty})(\Pi_{j=\ell+1}^{L-1}\|(\mathbf{W}^j)^T\|_{1,\infty}) + \mathbb{1}(L-1 \in I)\epsilon_{L-1}(\Pi_{i=1}^{L-2}\|\mathbf{W}^{i^*}\|_{1,\infty})\} \|\boldsymbol{x}_i\|_1$$

$$+ 2\mathbb{1}(L \in I)\epsilon_L(\Pi_{i=1}^{L-1}\|\mathbf{W}^{i^*}\|_{1,\infty}) \|\boldsymbol{x}_i\|_1\Big) + \frac{1}{\gamma}\Big(\frac{4}{n^{3/2}} + \frac{60\log(n)\log(2d_{max})}{n}\|X\|_F$$

$$+ \frac{2}{n} \sup_{f \in \mathcal{F}} \max_{k \in [K]} \left\{ \left\| W_{k,:}^L \right\|_1 \left\{ \sum_{\ell \in I \setminus \{L, L-1\}} \epsilon_\ell \left( \Pi_{i=1}^{l-1} \left\| \boldsymbol{W}^{i^*} \right\|_{1,\infty} \right) \left( \Pi_{j=\ell+1}^{L-1} \left\| (\boldsymbol{W}^j)^T \right\|_{1,\infty} \right) \right. \right.$$

$$\left. \left. + \mathbb{1}(L-1 \in I) \epsilon_{L-1} \left( \Pi_{i=1}^{L-2} \left\| \boldsymbol{W}^{i^*} \right\|_{1,\infty} \right) \right\} + 2\mathbb{1}(L \in I) \epsilon_L \left( \Pi_{i=1}^{L-1} \left\| \boldsymbol{W}^{i^*} \right\|_{1,\infty} \right) \right\} \left\| \boldsymbol{X} \right\|_{1,2} \right) + 3\sqrt{\frac{\log \frac{2}{\delta}}{2n}}$$

$$(65)$$

# D    Additional Experiments

## D.1    Different Settings of Figure 1 (a)

Figure 2 presents the empirical generalization gaps of different $\epsilon_{\text{train}}$ without weight PGD attack ($\epsilon_{\text{test}} = 0$). The generalization gap is obviously lower than others when $\epsilon_{\text{train}} = 0$, but its trend with respect to regularization is similar to the setting when $\epsilon_{\text{train}} \neq 0$.

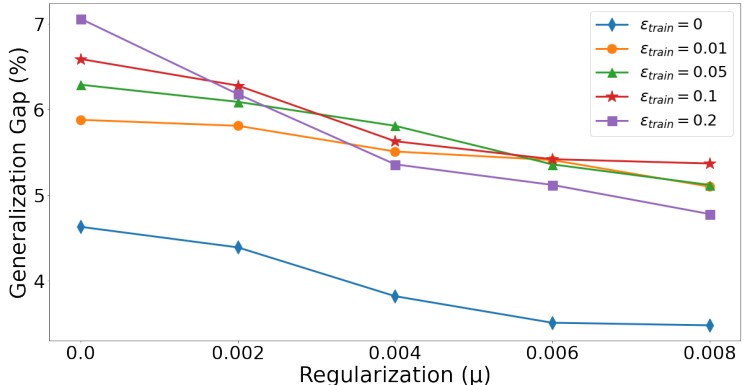

Figure 2: Empirical generalization gaps when altering the matrix norm regularization coefficient $\mu$ in (2) without weight PGD attack.

## D.2    Supplement of Figure 1 (a)

In Table 1, following same experiments of Figure 1 (a), we supplement test accuries under weight PGD attack with perturbation level $\epsilon_{\text{test}}$.

Table 1: Comparison of test accuracies of neural networks trained with different $\epsilon_{\text{train}}$ under weight PGD attack with perturbation levels $\epsilon_{\text{test}}$.

| $\epsilon_{\text{test}} = 0.001$ | | | | | |
|---|---|---|---|---|---|
| $\mu$ | 0 | 0.002 | 0.004 | 0.006 | 0.008 |
| $\epsilon_{\text{train}} = 0.01$ | 67.93% | 72.89% | 75.10% | 71.97% | 72.47% |
| $\epsilon_{\text{train}} = 0.05$ | 67.95% | 68.59% | 68.16% | 67.72% | 67.07% |
| $\epsilon_{\text{test}} = 0.002$ | | | | | |
| $\mu$ | 0 | 0.002 | 0.004 | 0.006 | 0.008 |
| $\epsilon_{\text{train}} = 0.01$ | 55.11% | 57.70% | 61.28% | 54.4% | 58.86% |
| $\epsilon_{\text{train}} = 0.05$ | 55.49% | 53.43% | 54.34% | 54.52% | 54.47% |

## D.3 Alternative Loss function of Regularization Term

To show the importance of our loss, in Table 2 and Table 3, we use $L_1$ and $L_2$ weight decay as regularizers, respectively. As shown in Table 2, $L_1$ weight decay can only endure regularization coefficient from $\mu = 0.0$ to $\mu = 0.002$ with $\epsilon_{\text{train}} = 0.01$. If $\mu$ is greater than 0.002, the training accuracy will be lower than 10%. Similarly, for the setting of $\epsilon_{\text{train}} = 0.05$, the training accuracy lower than 10% when $\mu$ is greater than 0. In Table 3, it presents that $L_2$ weight decay technique cannot reduce generalization gap as $\mu$ growing. Compared with results of our loss shown in Table 1, our loss either inhibits the generalization gap or makes models be more robust against weight PGD attack.

Table 2: Comparison of generalization gap and test accuracies of neural networks trained with $L_1$ weight decay and different $\epsilon_{\text{train}}$ under weight PGD attack with perturbation levels $\epsilon_{\text{test}}$.

| | | | | | |
|---|---|---|---|---|---|
| $L_1$ $\epsilon_{\text{test}} = 0.001$ | | | | | |
| $\mu$ | 0 | 0.002 | 0.004 | 0.006 | 0.008 |
| $\epsilon_{\text{train}} = 0.01$ | 5.13%/68.57% | 1.82%/53.58% | Failed to Converge | Failed to Converge | Failed to Converge |
| $\epsilon_{\text{train}} = 0.05$ | 5.16%/38.34% | Failed to Converge | Failed to Converge | Failed to Converge | Failed to Converge |
| $L_1$ $\epsilon_{\text{test}} = 0.002$ | | | | | |
| $\mu$ | 0 | 0.002 | 0.004 | 0.006 | 0.008 |
| $\epsilon_{\text{train}} = 0.01$ | 5.99%/43.5% | 2.42%/41.08% | Failed to Converge | Failed to Converge | Failed to Converge |
| $\epsilon_{\text{train}} = 0.05$ | 3.37%/27.83% | Failed to Converge | Failed to Converge | Failed to Converge | Failed to Converge |

Table 3: Comparison of generalization gap and test accuracies of neural networks trained with $L_2$ weight decay and different $\epsilon_{\text{train}}$ under weight PGD attack with perturbation levels $\epsilon_{\text{test}}$.

| | | | | | |
|---|---|---|---|---|---|
| $L_2$ $\epsilon_{\text{test}} = 0.001$ | | | | | |
| $\mu$ | 0 | 0.002 | 0.004 | 0.006 | 0.008 |
| $\epsilon_{\text{train}} = 0.01$ | 5.87%/68.13% | 3.58%/74.72% | 0.85%/71.55% | 5.08%/68.92% | 5.51%/63.72% |
| $\epsilon_{\text{train}} = 0.05$ | 4.12%/65.78% | 4.33%/ 60.97% | 3.6%/ 68% | 8.64%/57.56% | 8.63%/63.77% |
| $L_2$ $\epsilon_{\text{test}} = 0.002$ | | | | | |
| $\mu$ | 0 | 0.002 | 0.004 | 0.006 | 0.008 |
| $\epsilon_{\text{train}} = 0.01$ | 6.05%/52.85% | 3.31%/58.09% | 0.47%/57.47% | 5.09%/47.41% | 2.62%/42.68% |
| $\epsilon_{\text{train}} = 0.05$ | 6.82%/47.58% | 4.54% / 47.46% | 5.05%/51.05% | 5.7%/42% | 7.29%/47.79% |

## D.4 Convolutional Model Under PGD Weight Attack

We've also conducted the experiment of convolution-layer based model training on CIFAR-10 using our proposed loss function in Table 4. The model is primarily based on convolution layers and dense layers, with two convolution layers (32 filters of 5x5, and 64 filters of 5x5 in the second), three dense layers (128, 64, 10), ReLU activation and trained with $\epsilon_{\text{train}} = 0.01$ as prediction model. As shown in Table 4, the standard model's performance rapidly degrades in 5-folds when the perturbation radius is set to 0.001 while our model still retains half of its accuracy when compared to the no-perturbation setting ($\epsilon_{\text{test}} = 0$).

Table 4: Comparison of test accuracy of convolutional neural networks trained with $\epsilon_{\text{train}} = 0.01$ and different $\lambda$ and $\mu$ under weight PGD attack (200 steps) with perturbation level $\epsilon_{\text{test}}$.

| Parameters \ Perturbation Radius $\epsilon_{\text{test}}$ | 0 | 0.001 | 0.0015 | 0.0017 | 0.0019 | 0.0021 | 0.0023 | 0.0025 |
|---|---|---|---|---|---|---|---|---|
| $\lambda = \mu = 0$ | 58.73% | 13.08% | 10.15% | 10.01% | 10% | 10% | 10% | 10% |
| $\lambda = 3.24 * 10^{-4},\ \mu = 10^{-4}$ | 51.1% | 33.9% | 23.16% | 19.71% | 16.85% | 14.25% | 12.52% | 11.58% |
| $\lambda = 4.5 * 10^{-4},\ \mu = 10^{-4}$ | 47.28% | 36.82% | 27.78% | 26.19% | 21.6% | 19.35% | 17.84% | 16.53% |
| $\lambda = 5 * 10^{-4},\ \mu = 10^{-4}$ | 42.71% | 38.68% | 30.2% | 22.09% | 19.41% | 17.32% | 15.68% | 11.5% |
| $\lambda = 5.5 * 10^{-4},\ \mu = 10^{-4}$ | 42.79% | 32.9% | 25.3% | 22.77% | 20.78% | 18.88% | 18.26% | 17.1% |
| $\lambda = 6 * 10^{-4},\ \mu = 10^{-4}$ | 38.17% | 30.21% | 24.55% | 23.25% | 21.97% | 20.68% | 19.14% | 17.55% |

### D.5 Weight PGD Attack (100 steps) on MNIST

Figure 3 shows the accuracy and AUC score of different models against weight PGD attack with 100 iterations.

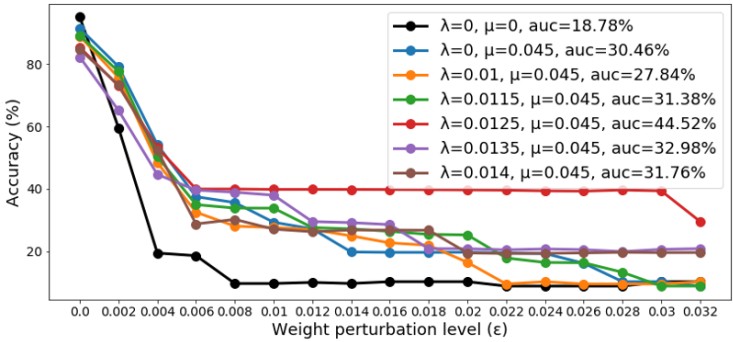

Figure 3: Comparison of test accuracy of neural networks trained with different coefficients $\lambda$ and $\mu$ against weight PGD attack (100 steps) with perturbation level $\epsilon$.

### D.6 More details on the trade-off between $\lambda$ and $\mu$

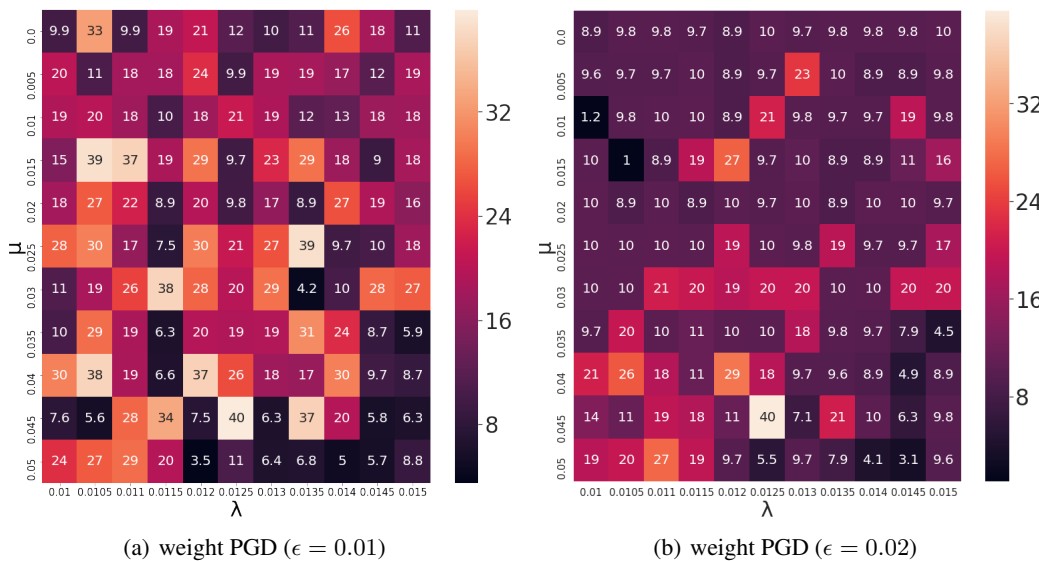

Figure 4: Comparison of test accuracy (%) trained with different coefficients $\lambda$ and $\mu$ against weight PGD attack (200 steps) with $\epsilon = 0.01$ and $0.02$. The experiment setting follows Figure 1(b).

Figure 4 shows the trade-off with a fine grid search between coefficients $\lambda$ and $\mu$, we present test accuracy under weight PGD attack using perturbation radius of ($\epsilon = 0.01$ and $0.02$) with different combinations of $\lambda$ (from 0.01 to 0.015) and $\mu$ (from 0 to 0.05). We find that there is indeed a sweet spot with proper values of $\lambda$ and $\mu$ leading to significantly better robust accuracy. When both $\lambda$ and $\mu$ are too large or too small, the robustness of the model will decrease.

### D.7 Alternative Robust Loss Function

In addition to the proposed loss function in equation (2), one can consider an alternative generalization gap regularization term derived from Theorem 4, which is $\sum_{m=1}^{L}(\log\left(||(\mathbf{W}^m)^\top||_{1,\infty}\right) + \log\left(||\mathbf{W}^m||_{1,\infty}\right))$. We compare its performance following the same experiment setting as in Figure

1 (b) with finetuned coefficients $\lambda$ and $\mu$. It can be observed that this alternative loss function also yields robust models with comparable (sometimes slightly better) performance to those in Figure 1 (b), verifying the effectiveness in using theory-driven insights to reduce the generalization gap against weight perturbation.

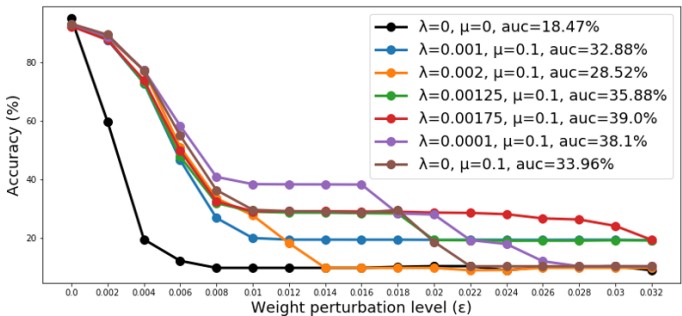

Figure 5: Test accuracy under different weight perturbation level $\epsilon$ with 200 attack steps. The models are trained using the alternative loss described in Section D.7.

## D.8 Run-time analysis

Table 5 reports the per-epoch run time of the models trained with the standard loss ($\lambda = 0, \mu = 0$) and the robust loss ($\lambda = 0.01, \mu = 0.1$) using equation (2). We train both models with 20 epochs and use the same hyperparameters, including setting the SGD optimizer with learning rate as 0.01, and setting the batch size as 32. The run-time cost of training both models are comparable.

Table 5: Per-epoch run time (in seconds) averaged over 20 epochs with the same hyperparameters for the standard ($\lambda = 0, \mu = 0$) and robust ($\lambda = 0.01, \mu = 0.01$) models based on equation (2).

|  | Standard model | Robust model |
| --- | --- | --- |
| Per-epoch run time | 5.125 sec. | 6.69 sec. |

## D.9 Robustness to Weight Quantization

We perform bit quantization on the weights of standard and robust models (the latter is trained using our proposed loss) with 32-bit training, and we report the corresponding test accuracy under different bit quantization levels. In Figure 6, one can observe that under 2-bit quantization, the accuracy of the standard model drops by 85.14%, whereas the robust model (green bar) only degrades by 28.71%, which is roughly 3x better than the standard model.

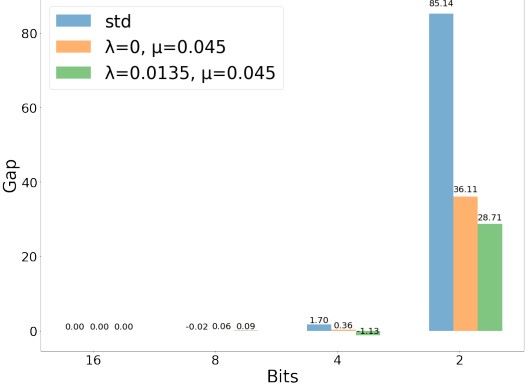

Figure 6: Comparison of the model performance with different quantization levels. The gap indicates the original test accuracy (32-bit) minus quantized test accuracy. Larger gap means worse robustness.

# E    On the Vacuity of Generalization Bound

In [14], empirical observations were made to point out the fact that when given increase in the size of the training data, the error bound proposed in [4] grows rapidly, loosing the ability to describe generalization gap and thus becomes vacuous. However, we note that under our settings with models trained using the loss function in Section 3.5, the bound would not grow in a polynomial rate and instead shows a decreasing trend. We conducted experiments and presented results under the same setting as [14] in Figure 7. Here we verify two existing generalization bounds from different literature, one from [4] while another one from [3] in which the former one is composed mainly of product of weight matrices' norm and the latter is comprised of the norm of matrices' product. Empirical results in Figure 7(a) show that under the standard settings the main components of generalization bound in [4] and [3] both grows rapidly with respect to the increase in size of the training dataset, as confirmed in [14]. Another empirical finding in the last column of Figure 7(a) shows that the multiplicative difference between bounds in [4] and [3] exhibit a constant rate, demonstrating the vacuity of both bounds. However, when measuring the same component under our setting in Figure 7(b), new results showed decreasing bounds as the size of the training dataset increases, concluding the non-vacuity of the associated generalization bounds in our settings.

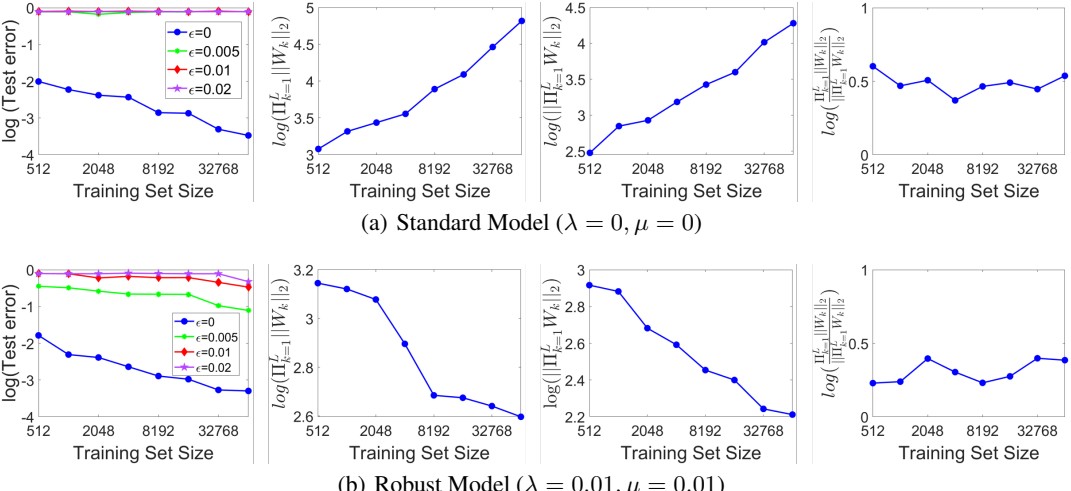

Figure 7: Statistics associated with generalization bounds for standard model and robust model trained using equation (2). The experiment setting follows Figure 1(a). In the first column, we present test error under weight PGD attack with perturbation level $\epsilon$. The curve with $\epsilon = 0$ (no weight perturbation) corresponds to the standard generalization setting. In the second column, we present the product of spectral norms of the weights matrices, related to bounds in [4]. In the third column, we show spectral norm of the product of the weight matrices, related to bounds in [3]. In the fourth column, we show the product of spectral norms of the weights matrices divided by the spectral norm of the product of the weight matrices. Notably, we present each value into the logarithm function. For the standard model, both types of bound increase with respect to training set size and are shown to be vacuous, consistent with the results in [14]. For the robust model, both types of bound exhibit same decreasing trend as the test error and therefore are non-vacuous. Moreover, these two bounds demonstrate similar scaling behavior (nearly constant log ratio) in both standard and robust models.

As an ablation study, Figure 8 shows the performance of the robust model with ($\lambda = 0.01, \mu = 0$). That is, training a neural network without the generalization regularization term in equation (2). Unlike the standard model, it is observed that the generalization bounds still show decreasing trend with the test error. This is due to the fact that the robustness term in equation (2), which is induced from our analysis of the worst-case error propagation from weight perturbation, also plays a role in regularizing the network weights, and therefore making the resulting generalization bound non-vacuous.

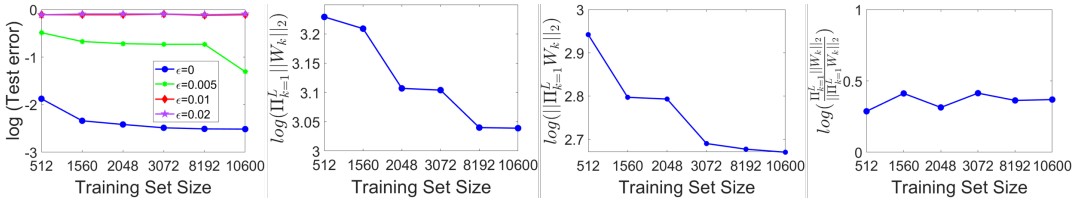

Figure 8: Ablation study of the robust model with ($\lambda = 0.01, \mu = 0$). The experiment setting is the same as Figure 7. In the absence of the generalization regularization term in equation (2), the generalization bounds still show decreasing trend with the test error.

# F  Weight Distribution of Each Layer for Standard and Robust Models

Figure 9 compares the weight distribution of each layer for standard and robust models. For both models, each layer has the mean value of weights close to zero. Moreover, both models tend to have larger weight variation for deeper layers. The weight distribution of robust model is observed to be more concentrated than that of standard model.

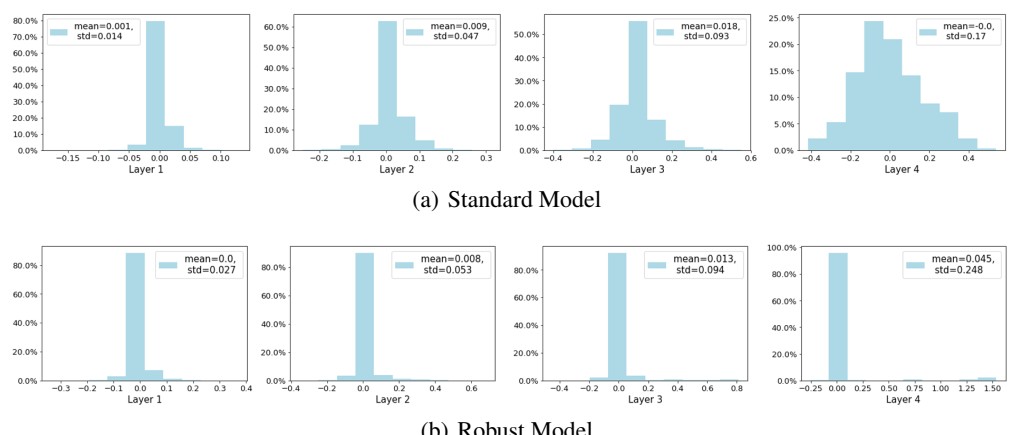

Figure 9: Visual comparison of weights distribution of each layer for the standard model with ($\lambda = 0, \mu = 0$) and the robust model with ($\lambda = 0.0125, \mu = 0.045$). The experiment setting is the same as Section 4.

## G   Comparison to Naive Adversarial Weight Training

As discussed in Section 3.4.1, we've explained why the naive weight adversarial training is intractable and invalid in notion. We present robust accuracy of the suggested weight adversarial training using the MNIST dataset shown in Table 6. The experimental setup follows that of Figure 1(b) and set $\epsilon_{\text{train}}$ as 0.01. As shown in Table 6, using the naive weight adversarial training provides little utility contrasted to models trained under our proposed loss.

Table 6: Comparison of test accuracies of neural networks trained by different methods such as weight adversarial training, our loss and standard training under weight PGD attack with different perturbation radius $\epsilon_{\text{test}}$. Weight adversarial training and our loss are trained with $\epsilon_{\text{train}} = 0.01$.

| Perturbation Radius $\epsilon_{\text{test}}$ | 0.001 | 0.005 | 0.008 | 0.01 | 0.15 | 0.02 |
|---|---|---|---|---|---|---|
| Weight AT Acc | 10% | 12% | 14% | 10% | 14% | 12% |
| Ours Acc | 80.3% | 50.2% | 39.87% | 39.83% | 39.8% | 39.57% |
| Standard Acc | 70.32% | 19.01% | 9.74% | 9.74% | 8.92% | 8.92% |

## H   Discussion of References

Table 7: Comparison of key differences between our paper and other references.

| | Generalization Settings | Types of Robustness | Measure of Generalization Gap | Methods of Evaluating Maximum(Worst Case Loss) |
|---|---|---|---|---|
| Ours | Robust Setting | Against **Weight** | $\mathbb{P}[\exists (w+\epsilon) s.t. \mathbb{1}(y \neq argmax[f_{w+\epsilon}(x)]_{y'})]$ $\leq \frac{1}{n}\Sigma \mathbb{1}(M(f_w(x), y) - \psi(f_w(x) \neq \gamma) + \text{Gap}$ | Layer Propagation (Exact Bound) |
| Adversarial Weight Perturbation (2020) | Robust Setting | Against Input | $L_{D,\epsilon}(w+\epsilon) \neq L_{S,\epsilon}(w+\epsilon) + \text{Gap [see eq.12]}$ | Maximum value based on generated adversarial inputs (inexact bound) |
| Sharpness-Aware Minimization (2021) | Standard Setting | NA | $L_D(w) \neq max_\epsilon L_S(w+\epsilon) + \text{Gap [see Appendix A.1]}$ | First-Order Taylor Expansion (approximation; inexact bound) |