# OpenReview forum: "Formalizing Generalization and Adversarial Robustness of Neural Networks to Weight Perturbations"
_NeurIPS.cc/2021/Conference — NeurIPS 2021 Poster_

### Official Review · Reviewer_LdZY · 2021-07-13

**Rating:** 7
**Confidence:** 4

**Summary:**

In this work the authors study the propagating effect of perturbing the weights of a neural network on the models final output. They use the resulting theorems to construct a bound on the Rademacher complexity of the model class under perturbed weights and provide regularizations terms to improve both the robustness against these weight perturbations and to reduce the generalization gap.

**Limitations And Societal Impact:**

As I stated in my main review, the numerical evaluation (and more specifically the experimental setup) leave me to question a few practical implications of the theory presented. The authors should clarify these in their reply.

**Main Review:**

Dear authors and AC,

Below you will find my review for the paper in question.

As very clearly laid out in the introduction, the study of a neural networks sensitivity to perturbations of its weights is an research area of practical importance. The first theoretical characterization provided in this present work is therefore of considerable importance and might have significant impact on future work. I must however admit that I am not entirely up to date with research regarding weight quantization and compression. For this reason I would like to leave judging the works novelty to the other reviewers.

My main concerns however lie with the numerical validation performed and the unanswered questions regarding the practical usefulness of the bounds and regularization terms that arise from this section.
- First and foremost, the experimental setup suggest that the additional of the surrogate loss might bring some serious computational restrictions. As far as I understand the experiments have been performed with only 1000 MNIST data samples for training. In the manuscript there is no explanation offered for this decision besides simply following related work. I would appreciate it if the authors could comment on this and/or provide some additional data showcasing the computational cost of training the model using the surrogate loss. The discussion offered in appendix section D.7 is in my opinion not clear enough. (eg. equation 2 referred to in this section seems to only take first layer perturbations into account)
- Second, in the experimental setup the authors mention that no bias terms were used. It would be good to clarify why these were excluded. And, if this is because of them not fitting in the framework, this limitation should be more clearly communicated.
- Third, I have my reservations regarding the effectiveness of the evaluation metric used. Specifically, I do not understand the use of the AUC metric. Even the best performing model in figure 1b quickly degrades to roughly 40% accuracy. At this level of accuracy the usability of the models is mostly completely gone. The further slow degradation after is therefore not really informative any more. In similar fashion, the supplementary experiment on weight quantization shows that the decrease in accuracy of a robust model is 3x less then a standard model for 2 bit quantization. However, this still degrades the model accuracy far enough to question its use. For 16-4 bit quantization the standard and robust model perform roughly the same.
- Lastly, I believe that a comparison against standard L1 and L2 weight regularization is somewhat necessary. I suspect that keeping the norm of the weights small should suffice to see a lot of the same benefits as the proposed regularization terms.

Despite the shortcomings of the numerical evaluation section mentioned above, I still believe that the theoretical study presented in the paper can be a valuable starting point for further studies. For this reason I am still inclined to vote for acceptance. If the concerns with the numerical evaluation above could be expanded upon by the authors I would be willing to improve my score.

Some other minor remarks:
- I appreciate the use of colour to clarify the mathematical notation, but I don't think it is used very effectively in some cases. For example, I do not see the use of colouring the equation in sec. 3.5.
- There are numerous references to equation 2. These are broken and very  confusing.
- Line 308 has a dead reference.

**Time Spent Reviewing:**

5 hours

---

> ### Author Response · Authors · 2021-08-10
> **Response to Reviewer LdZY**
>
> We would like to first thank you for the recognition and endorsement of our work! We are delighted to receive the positive feedback that *“The study …. is a research area of practical importance.”* and *“The first theoretical characterization provided in this present work is therefore of considerable importance and might have significant impact on future work.”*
>
> Here we would address doubts raised in the reviews by points as follows.
>
> &nbsp;
> ### Regarding Data Usage on the Generalization Gap
>
> ---------------
>
> We would like first to apologize for not lucidly characterizing the experiment and relevant data usage. The purpose of using only 1000 MNIST data points is to showcase and validate the phenomenon and theory (Theorem 4 in the paper) that models trained against weight perturbation would suffer from the widening of generalization gap, instead of a limitation of scalability. As mentioned in Line 308 (with ?? being replaced by [R1]), this setup follows the setting in [R1] to study the widened generalization gap under input perturbations. For completeness, for a full-scale dataset, we’ve presented the study of robustness in Figure 1(b).
>
> We’ve also included convolution model training on CIFAR in the table below. The model is primarily based on convolution layers and dense layers, with two convolution layers (32 filters of 5X5, and 64 filters of 5X5 in the second), three dense layers (128, 64, 10) and ReLU activation as prediction model.
>
> &nbsp;
> #### Convolutional Model  PGD Weight attack
> ---------------
> |              Perturbation Radius $\epsilon$               |   0    | 0.001  | 0.0015 | 0.0017 | 0.0019 | 0.0021 | 0.0023 | 0.0025 |
> |:-------------------------------------:|:------:|:------:|:------:|:------:|:------:|:------:|:------:|:------:|
> |         $\lambda=0, \mu=0$          | 58.73% | 13.08% | 10.15% | 10.01% |  10%   |  10%   |  10%   |  10%   |
> | $\lambda = 3.2*10^{-4}, \mu=10^{-4}$ |    51.1% | 33.9%  |23.16% |19.71% | 16.85%  | 14.25% |  12.52%      |11.58%        |
> | $\lambda=4.5*10^{-4}, \mu=10^{-4}$  | 47.28% | 36.82% | 27.78% | 26.19% | 21.6%  | 19.35% | 17.84% | 16.53% |
> |  $\lambda = 5*10^{-4}, \mu=10^{-4}$  | 42.71% | 38.68% | 30.2%  | 22.09% | 19.41% | 17.32% | 15.68% | 11.5%  |
> |         $\lambda=5.5*10^{-4}, \mu=10^{-4}$         | 42.79% | 32.9%  | 25.3%  | 22.77% | 20.78% | 18.88% | 18.26% | 17.1%  |
> |   $\lambda=6*10^{-4}, \mu=10^{-4}$   | 38.17% | 30.21% | 24.55% | 23.25% | 21.97% | 20.68% | 19.14% | 17.55% |
>
>
> As one can see, the standard model’s performance rapidly degrades in 5-folds when the perturbation radius is set to 0.001 while our model still retains half of its accuracy when compared to the no-perturbation case.
>
> Furthermore, we thank you for pointing out the error of reference in section D.7. The run-time analysis is conducted under the scheme of all-layer perturbations which refers to Theorem 2 instead of equation 2 in the paper.
>
> &nbsp;
> ### Regarding Bias in the Framework
> ---------------
>
> We thank you for raising the question on introducing bias into the whole framework. We will make a remark that putting bias into the framework does not affect the overall outcome of the theorems and results.
>
> Specifically, we could include the bias term into a part of the weight matrix to contrive the pairwise margin bound of $W^{\prime}$ and $x^{\prime}$ but with more sophisticated mathematical expressions (so we didn’t consider the bias term for mathematical simplicity), where $W^{\prime} = [W, b^T] $and $x ^{\prime}= [x, 1]^T$. Furthermore, if you would like to see the additional experiment on this, we are willing to include this as an additional result.
>
> &nbsp;
> ### Regarding the Use of Metric and Supplementary Experiments
> ---------------
>
> We appreciate for bringing up the discussion on the use of standard metrics of AUC. The AUC metric is intuitive because for an ideal robust model, the AUC score would be at its maximal value, meaning the model attains perfect accuracy across the spectrum of the tested weight perturbations. Here we would also like to emphasize that the standard model in Fig 1(b) degrades heavily to 20% even when the perturbation radius is merely 0.004 where in contrast most of our models retain accuracy of above 50% with the best accuracy at 80%.
>
> On the other hand, we note that a similar trend persists in the realm of input perturbation where PGD attacks after many iterations could also lead to devastating performance. From a different perspective, this highlighted the use of generalization bound under perturbation in which we could more accurately assess the overall performance of the model.
>
> Nonetheless, we hope that this paper could spark a commencement in the research area of robust training against parameter perturbation which other than security reasons also possess many uses in relevant areas (such as those included in introduction).
>
> &nbsp;
> ### Comparison of $L_1$/$L_2$ Regularizations
> ---------------
>
> Thanks for your valuable insight in the constraint of weights regarding generalization. We would like to note that although these different norms all serve the purpose of reducing weight value, our regularization term acts in accordance with the proposed scenario (according to Theorem 4) and is the smallest of these techniques.
>
> Furthermore, the comparison to $L_1$ regularization was already presented in Appendix D.2, where we show our regularizer performs better. As an extension to our experiment conducted in D.2 regarding different forms of regularization, we’ve included the result of $L_2$ regularization below, which shows that these regularization methods can all serve to be one possible candidate in controlling the weights magnitude; however, using our loss the generalization gap is further reduced and can better reflect model’s performance.
>
> &nbsp;
> #### Generalization Gap (%) /Test accuracy (with  $\epsilon= 0.01$)
>
> ---------------
>
> |  Coefficient $\mu$   |   0   | 0.002 | 0.004 | 0.006 | 0.008 |
> |:--------:|:-----:|:-----:|:-----:|:-----:|:-----:|
> |  $L_1$   |  6.36% / 84.2% |  6.09% / 84.8% |   Failed to Converge  |   Failed to Converge   |   Failed to Converge  |
> |  $L_2$   | 6.26% / 84.3% | 6.1% / 83.2%| 5.78% / 84.5%  | 5.6% / 78.8%|  5.5% / 78.5% |
> | Our Loss |  5.88% / 85.8% | 5.81% / 85%  | 5.51% / 87.8% |  5.41% / 83.4% | 5.1% / 85.1% |
>
> &nbsp;
> ### On Minor Remarks
> ---------------
> We appreciate the remarks offered in the review and are sorry if the color usage has caused any misunderstanding. Our original purpose of coloring in Section 3.5 is to utilize colors to link different parts of the proposed loss functions to different parts of the theorems. We will use additional underbrackets to make their correspondence more clear.
>
>  Secondly, any references to equation 2 is actually mainly pointing to the loss function in Sec 3.5 (Line 296 - 297) which is the all-perturbed scenario. We apologize for the referencing error. Furthermore , Line 308 refers to the reference [R1], which is not properly compiled but we would fix them immediately as you suggested.
>
> ---------------
>
> [R1] Yin, D., R. Kannan, P. Bartlett. Rademacher complexity for adversarially robust generalization. In International Conference on Machine Learning, pages 7085–7094. 2019.

---

> > ### Comment · Reviewer_LdZY · 2021-08-24
> > **Thank you for the clarifications.**
> >
> > Thank you for the clarifications. Based on the additional experimental results I more strongly believe that the paper would be a worthwhile contribution to the conference.

---

> > > ### Author Response · Authors · 2021-08-24
> > > **Thank you for the encouraging comment and score increase**
> > >
> > > We thank the reviewer for the encouraging comment "I more strongly believe that the paper would be a worthwhile contribution to the conference." and for your constructive comments. We are also delighted that the reviewer increases the review rating.

---

### Official Review · Reviewer_Ljmn · 2021-07-15

**Rating:** 6
**Confidence:** 4

**Summary:**

In this paper, the authors developed a formal analysis of the robustness associated with the pairwise class margin for neural networks against weight perturbations. The authors also characterized its generalization gap through Rademacher complexity and proposed a theory-driven loss function for robust generalization. The empirical results showed significantly improved performance in generalization and robustness.

**Limitations And Societal Impact:**

see above

**Main Review:**

1. The authors aim to study a new problem in neural networks’ robustness to weight perturbations. While the concept is kind of similar to adversarial examples, I would love to see some explanation on why the weight perturbation adopts the Linf norm constraint by default.
2. One of the major issues in the paper is that it lacks necessary comments and intuitions for each theoretical result. For example, what does theorem 1 implies? Or how does this result help us? It is not clearly written. Readers need to go over the entire paper to understand the author's purpose to derive such an upper bound.
3. Theorem 3, what is the Error of Weight Perturbation term defined? Should make it clear in the main paper
4. Until Section 3.5, I start to understand that the authors are trying to propose a new surrogate loss for weight robust training. However, the question is: is this surrogate loss necessary? In the experiments, the authors only compare their own method in different parameter settings. However, a simple baseline to consider here is simple weight adversarial training (just like PGD attack to weight PGD attack, it is easy to build adversarial training to weight adversarial training). In this way, it does not require any surrogate loss derived in this paper. In other words, it makes much more sense in the case that weight adversarial training is performing poorly and the authors turn to study other surrogate losses for this problem. Yet if weight adversarial training is already doing ok, I don’t see the reason for proposing such a surrogate loss.


Minor comments:

1 .   Line 308, typo (??)


**Time Spent Reviewing:**

3

---

> ### Author Response · Authors · 2021-08-10
> **Response to Reviewer Ljmn**
>
> Thank you for your sincere suggestions and valuable feedback! We are delighted to receive the positive feedback that *“The authors aim to study a new problem in neural networks’ robustness to weight perturbations.”*
>
> We are truly content that you appreciated the work and have raised several questions which we will answer by points below.
> &nbsp;
> ### The Notion of $L_\infty$ Norm Constraints
> _______
>
> As raised in the introduction (line 20 - 22), the usage of $L_\infty$  constraints on weights perturbation arises in multiple scenarios such as investigating the quantized neural networks, fault injection attack (using row hammer to flip bits in memory) of any given neural network and the study of bit error robustness of DNN accelerators. The $L_\infty$  constraint entails the physical meaning of maximal perturbation allowed on each parameter of the model weights.
>
> &nbsp;
> ### Regarding Implications and Notation of the Theorems
>
> _____
>
> Thanks for pointing out the crucial point, we will improve the notation and add more discussion of the implications in the revised version. Below we summarize the presentation flow of this work followed by the order of theoretical statements and their implications.
>
> Firstly, we would like to provide a description here on all the derived theorems and lemmas as below. As stated in the paper, the direct optimization of the min-max loop simultaneously operated on in the parameter space is not viable (see the following section for more information) and therefore we aim to devise a surrogate loss to train against weight perturbations by first developing bounds on the pairwise margin difference between standard and perturbed scenarios. In Theorem 1, we consider the case of single layer perturbation, and develop the notion of bound propagation. We then utilize these as the building block to further construct the scenario when all-layers are perturbed or only partial layers are perturbed, which is presented in Theorem 2 and Theorem 3.
>
> Furthermore, we also investigated the generalization gap of models trained under this surrogate loss and discovered in Theorem 4 that the generalization gap, having dependence on the weight norm, would widen if not properly controlled. As a consequence, we proposed a theory-driven loss in Sec 3.5 to mitigate this effect.
>
> &nbsp;
> ### Regarding the Definition of the Error Term
> ___
>
> With regard to the error of weight perturbation in Theorem 3, it is defined as the sum of the perturbed layer error (marked in light blue) and the final layer error (marked in light brown). We will make it more specific in the revised version.
>
> &nbsp;
> ### Explanations of Adversarial Weight Training
> ___
>
> We thank you for mentioning this issue. As discussed in Section 3.4(Line 240 - 242), we’ve explained why the naive weight adversarial training is intractable and invalid in notion. We present here a table of robust accuracy of the suggested weight adversarial training using the MNIST dataset as below. The experimental setup follows from result Figure 1.(b) in the paper.
>
> &nbsp;
> #### MNIST Dataset
> _____
>
> | Perturbation Radius $\epsilon$ | 0.001 | 0.005 | 0.008 | 0.01 | 0.15 | 0.02 |
> | ---------- |:-----:|:-----:|:-----:|:----:|:----:|:----:|
> |  Weight AT Acc        |  10%  |  12%  |  14%  | 10%  | 14%  | 12%  |
> | Ours Acc |   80.3%       | 50.2%          | 39.87%        | 39.83%         |    39.8%      |    39.57%      |
> | Standard Acc |    70.32%      |   19.01%    |      9.74%      |   9.74%       |     8.92%     | 8.92%      |
>
> As one can see, using the naive weight adversarial training provides little utility contrasted to models trained under our proposed loss. We believe this result further motivates our work and will include it in the revised version.

---

> > ### Comment · Reviewer_Ljmn · 2021-08-24
> > **Thank you**
> >
> > I thank the authors for the detailed response. After clarifying all those points, I would like to raise my score to 6. Although the presentation and numerical evaluation part certainly need many improvements (as mentioned by other reviewers), I am ok to see it accepted.

---

> > > ### Author Response · Authors · 2021-08-24
> > > **Thank you for your support and score increase**
> > >
> > > We thank the reviewer for increasing the review score and for the constructive comments. We agree with the reviewer that the additional numerical results reported in the rebuttal can further improve this work, and we will include them and the suggested changes in the revised version.

---

### Official Review · Reviewer_Yipw · 2021-07-15

**Rating:** 5
**Confidence:** 4

**Summary:**

The paper studies the effect of weight perturbations in a neural network on its performance in terms of generalization and adversarial robustness. Robustness to weight perturbations is important for improving a model’s fault tolerance and reducing its storage requirements without significantly compromising its performance. It uses Rademacher complexity to quantify the impact on generalization and pairwise class margins for adversarial robustness. It considers networks with non-negative monotone activation functions that are 1-Lipschitz. Using the theory developed in the paper, the authors propose a new loss function for training more generalizable and robust models.

**Limitations And Societal Impact:**

This paper seeks to improve the robustness and generalizability of neural network models. The only limitation that comes to my mind of this type of work is its negative impact on the accuracy of the model. The authors have studied the trade-off between robustness and accuracy. However, as I mentioned in the main review a study of the test accuracy of the trained model with respect to the loss parameters is important to properly understand the gain in generalization capabilities.

**Main Review:**

Originality: To the best of my knowledge, this work is the first study in understanding the effects of weight perturbations on a neural network’s ability to generalize on unseen data and its adversarial robustness. It combines well-known mathematical techniques to quantify and characterize a model’s robustness and generalization capability.

Quality: The theoretical results appear to be technically sound though not particularly surprising. The experiments study how the generalization gap and robust accuracy vary with the parameters of the proposed loss terms. However, it is not clear how they affect the test accuracy. Fig 1(a) shows that the gap between train and test accuracies decreases as mu increases for different values of epsilon used during training. Does this drop in the gap result in any gain in test accuracy? Is there a setting of the parameters that helps achieve a better test accuracy than when the proposed loss function is not used? I believe showing an improvement in the test accuracy will better demonstrate the gain in generalization from the proposed loss function.

Clarity: The paper is well written.

Significance: The authors demonstrate through experiments that the proposed loss function helps improve robustness and generalization which are important qualities for any good machine learning model. However, it is difficult to determine the significance of this work without knowing the effect of the loss function parameters on test accuracy. One can always construct models that have high robustness and low generalization gap but have poor accuracy. I believe a study of how these parameters affect the accuracy of the model would help increase the impact of this paper.

**Time Spent Reviewing:**

8

---

> ### Author Response · Authors · 2021-08-10
> **Response to Reviewer Yipw**
>
> Thanks for your genuine appreciation of the clarity and precious reviews on our work! We are delighted to receive the positive feedback that *“this work is the first study in understanding the effects of weight perturbations on a neural network’s ability to generalize on unseen data and its adversarial robustness.”*
>
> Please see our point-to-point response to your comments below.
>
> &nbsp;
> ### Experiments on the Relationship Between Robustness, Generalization Gap and Accuracy
> ___
>
> We thank you for indicating the issues between robustness, generalization gap and accuracy which we will address by parts below.
>
> #### **Experiment on Generalization Gap and Accuracy**
> Here we present the experiment of generalization gap and test accuracies with varying $\mu$ and $\epsilon$ following our training loss defined in Section 3.5. The experiment setup is the same as Figure 1 (a).
>
> &nbsp;
> #### MNIST (Generalization Gap (%) / Test Accuracy (%) )
> ___
>
> |     Coefficient  $\mu$      |   0   | 0.002 | 0.004 | 0.006 | 0.008 |
> |:---------------:|:-----:| ----- | ----- | ----- | ----- |
> | $\epsilon=0.01$ | 5.88% / 85.8% |   5.81% / 85%    | 5.51% / 87.8%      |       5.41% / 83.4%  | 5.1% / 85.1% |
> | $\epsilon=0.05$ |   6.29% / 83%    |   6.09% / 81.3%   |  5.81% / 80.3%  |  5.36% / 81.5%     | 5.12% / 81% |
> | $\epsilon=0.1$  |   6.59% / 76%    |  6.28% / 78.6%     |     5.63% / 75.9%  |    5.42% / 76.9%   |  5.37% / 79.5%     |
> | $\epsilon=0.2$  | 7.06% / 71.5%  | 6.18% / 66.5%  |    5.36% / 68.6 %   |    5.12% / 69.5 %   |  4.78% / 65.2%     |
>
> As one can see in the table above, if we speculated the results row-wise, one could spot that albeit the generalization gap exhibits a decreased trend, the test accuracy is not strictly increasing. Here we note that, this is due to the fact that as the coefficient $\mu$ increases, the target of the training model becomes more inclined to control the generalization gap as opposed to the standard accuracy metric.
>
> However, we see that the test accuracies are not of great deviation or deteriorate badly which in turn render the model useless. Instead, the model only presents a tolerable deviation within 5%. This indicator further signifies that our model could in turn balance the trade-off between accuracy and generalizability, better reflecting the model’s performance in the training process. The results also suggest that our loss (with proper regularization) will not lead to *models having high robustness and low generalization gap but poor accuracy*.
>
> #### **Experiment on Parameter Setting**
>
> We believe that the reviewer is asking the comparison to the setting when $\lambda  = \mu = 0$ based on the loss term in Section 3.5, which we already compared in Fig. 1(b). Here we note that given small perturbation the standard model would be heavily devastated. Meanwhile, under our setting there could be sweet spots of the parameter values of $\lambda$ and $\mu$ that could achieve much robust performance as enclosed in Appendix D.7

---

> > ### Comment · Reviewer_Yipw · 2021-08-24
> > **Comments on originality and questions about parameters**
> >
> > Thank you for the additional experiments!
> > Is the $\epsilon$ here the same as the one used during training (Fig 1 (a) in the paper) or the size of the adversarial perturbations (Fig 1 (b))?
> > Is the test accuracy reported in the above table the clean accuracy of the model (under no adversarial attack) or is it under the presence of a PGD attack of size $\epsilon$?
> >
> > Originality: I am not so sure anymore if this work can be called "the first study in understanding the effects of weight perturbations on a neural network’s ability to generalize on unseen data and its adversarial robustness" as I wrote in my initial review after coming across the references pointed out by one of the other reviewers which have studied these effects.

---

> > > ### Author Response · Authors · 2021-08-24
> > > **Thank you for your feedback and our follow-up clarifications**
> > >
> > > We thank the reviewer for the continued discussion. Please see our point-by-point follow-up response below.
> > >
> > > >Is the $\epsilon$ here the same as the one used during training (Fig 1 (a) in the paper) or the size of the adversarial perturbations (Fig 1 (b))?
> > >
> > > Ans: As we specified in our response, the experiment setup is the same as Figure 1 (a). Therefore, the $\epsilon$ here refers to the one used during training.
> > >
> > > >Is the test accuracy reported in the above table the clean accuracy of the model (under no adversarial attack) or is it under the presence of a PGD attack of size $\epsilon$?
> > >
> > > Ans: The reported test accuracy is the accuracy of the unperturbed model trained with our robust loss. We also observed a similar trend when the model is under adversarial weight perturbation. If the reviewer is interested, we can report these additional results. Please let us know.
> > >
> > > >Originality
> > >
> > > Ans: We understand the reviewer's concern but we would like to raise a different opinion. Comparing to the new references provided by other reviewers, we believe the discussion on their fundamental difference in problem scope actually further clarifies our contribution in studying and characterizing the generalization of neural networks under worst-case weight perturbations. For example, in our response to Reviewer sPbL regarding "Discussion of References", it is clear that other methods consider different problem settings than ours, either standard generalization or robust generalization against *input* perturbation, but none of them consider robust generalization against worst-case **weight** perturbations (our studied problem setting). We hope this clarification addresses the reviewer's concern.

---

> > > > ### Comment · Reviewer_Yipw · 2021-08-24
> > > > **Re: Thank you for your feedback and our follow-up clarifications**
> > > >
> > > > Thank you for the clarifications on the parameter $\epsilon$! I got confused between the two $\epsilon$ used for experiments reported in Fig. 1(a) and Fig. 1(b). It might be a good idea to denote the training $\epsilon$ and the adversarial perturbation size $\epsilon$ with different notations, perhaps with subscripts $t$ and $p$ respectively.
> > > >
> > > > My last comment regarding originality was just to correct my own statement in the initial review, which was misleading. I understand that this paper studies a different setting which technically makes it the first of its kind.

---

> > > > > ### Author Response · Authors · 2021-08-24
> > > > > **Thank you for your prompt response**
> > > > >
> > > > > We thank the reviewer for the prompt response. We will follow your suggestion to use two different notations for the perturbation levels used during training and testing. We are also glad that the comment on originality is clarified.
> > > > >
> > > > > We really appreciate your careful reading and constructive comments to improve the quality of this work. If there are any other suggestions or comments that you would like to make, please don't hesitate to let us know!

---

> ### Author Response · Authors · 2021-08-29
> **Additional Results and Our Words for Reviewer Yipw**
>
> ### Additional Experiments Based on Reviewer's Comments
> ___
>
> We thank the reviewer for the recent suggestion and response regarding the interest of inspecting the generalization gap and test accuracy under weight perturbation (see https://openreview.net/forum?id=hOG8swMRmY&noteId=FUQDfjhPP33).
>
> As a follow-up experiment, we present the additional generalization behavior and the performance of models trained against weight perturbation. We note that both train and test accuracies are measured under weight PGD attack. The table enclosed below concludes the result where $\epsilon_\text{train}$ is the perturbation radius under which the model was trained and $\epsilon_\text{test}$ represents the radius where train and test accuracies where measured.
>
> &nbsp;
>
> ### Generalization Gap (%) / Test Accuracy (%)
> ___
> **$\epsilon_\text{test} = 0.001$**
>
> |            $\mu$             |      0       | 0.002 | 0.004 | 0.006 | 0.008 |
> |:----------------------------:|:------------:| ----- | ----- | ----- | ----- |
> | $\epsilon_\text{train}=0.01$ | 3.57%/67.93% |  3.24%/72.89%     |   2.61%/75.10%    |   1.32%/71.97%    |   1.22%/72.47%    |
> | $\epsilon_\text{train}=0.05$ |      3.70%/67.95%        |   3.66%/68.59%    |   2.49%/68.16%    |  1.63%/67.72%     |    1.18%/67.07%   |
>
> &nbsp;
>
> **$\epsilon_\text{test} = 0.002$**
>
> |            $\mu$             |        0         | 0.002          | 0.004 | 0.006         | 0.008 |
> |:----------------------------:|:----------------:| -------------- | ----- | ------------- | ----- |
> | $\epsilon_\text{train}=0.01$ | 3.65%/55.11%  | 3.15%/57.70% |  1.57%/61.28%     | 1.20%/54.4% |   1.16%/58.86%    |
> | $\epsilon_\text{train}=0.05$ |7.11%/55.49% |        5.67%/53.43%        |   3.31%/54.34%    |           2.98%/54.52%    |  0.38%/54.47%     |
>
> As one inspect row-wise, when our coefficient $\mu$ increases, the observed generalization gap is as well decreased matching theoretical discovery which in turn offers us a better and accurate way to assess the model's performance under weight adversarial perturbation.
>
>
> &nbsp;
> ### Our Words for Reviewer
> _____
> In the end of the response, we would like to express sincere gratification towards the reviewer's effort in the overall review. We appreciate the reviewer for their invaluable time and indispensable contribution to the paper in which these inputs would definitely cast fresh lights on the refinement of the manuscript.
>
> We also would like to thank the reviewer for setting up positive tone from the beginning which is synergetic for the entire discussion. Overall, we've had a smooth and thought-provoking conversation throughout the rebuttal period which largely attributed to the calm mindset and expertise of the reviewer.
>
> **Lastly, we hope that as a whole our response and additional empirical evaluation would be helpful in your final evaluation.**

---

### Official Review · Reviewer_sPbL · 2021-07-21

**Rating:** 5
**Confidence:** 4

**Summary:**

The paper studies the worst-case perturbations in the weight-space for multi-layer networks. The authors first derive a robust surrogate loss which is an upper bound on the loss under norm-bounded weight perturbations based on a product of the 1-inf matrix norm of layer weight matrices. Then the authors derive a generalization bound based on the Rademacher complexity which motivates their new loss function which is evaluated empirically on MNIST.

**Limitations And Societal Impact:**

I think the authors could have discussed better the tightness of their bound on the worst-case loss. The authors don’t discuss any negative societal impact of their work but I think it’s not necessary for this paper.

**Main Review:**

**Overall assessment**:
- Originality: the derived bounds (both for the worst-case loss and for generalization) are to the best of my knowledge novel in the context of weight perturbations, although they seem to be directly inspired by similar bounds in the context of input perturbations.
- Quality: the submission seems to be technically sound but the empirical evaluation can be significantly improved (see **Cons**).
- Clarity: the paper is clearly written and well-organized.
- Significance: on the one hand, theoretically characterizing generalization is an important contribution but on the other hand I have concerns about the tightness of the proposed bound and usefulness of the proposed loss function (see **Cons**).

**Pros**:
- Formal analysis of generalization and the behaviour of the worst-case loss with respect to weight-space perturbations is a valuable contribution.
- Novel theory-driven loss function that aims at improving generalization and robustness against weight perturbations, although I have some concerns about its effectiveness.



**Cons**:
- My main concern is the tightness of the derived bounds on the worst-case loss. Analogous layerwise bounds are simple and known for a long time in the context of **input-space** perturbations (as opposed to the **weight-space** perturbations studied in this paper), e.g. see
[Certified Defenses against Adversarial Examples](https://arxiv.org/abs/1801.09344), Section 5 which can be trivially extended to multiple layers and to norms on weight matrices W^i other than the spectral one (e.g., including the 1-inf norm that this paper uses a lot). Although the loss function from Sec. 3.5 is not exactly the same as the objective that [Certified Defenses against Adversarial Examples](https://arxiv.org/abs/1801.09344) use as a baseline, they seem to be very connected since the proposed loss relies mainly on the 1-inf norm of the weights. Thus, it would be very useful if the authors could (1) discuss this connection and (2) empirically/theoretically justify the tightness of the proposed bound. The concern (2) comes from the fact that this layerwise bound is usually loose for the input-space perturbations, and thus has been used in the literature just as a baseline (and more involved approaches have been developed, e.g. the SDP relaxation of [Certified Defenses against Adversarial Examples](https://arxiv.org/abs/1801.09344)). There is an analysis of the tightness of the generalization bound in Appendix E but the tightness of the bounds on the worst-case loss doesn’t seem to be sufficiently discussed.
- The empirical evaluation part requires some natural baselines such as, e.g., standard L1/L2 weight regularization. It seems that both additional terms from the theory-driven loss (from Sec. 3.5) are based primarily on the 1-inf norm of the weights, so a natural question is how different it is from the standard weight regularization methods. I don’t think that the current experimental part really justifies the proposed method. Related to this, I don’t think that Fig. 3 in the appendix is informative enough since it shows just the generalization gap but what is more interesting is the absolute value of test accuracy.
- Another aspect of the empirical evaluation which is not very strong is that the evaluation is done only on one small-scale dataset (MNIST). As the authors note in the “Run-time analysis” paragraph, the method should be quite scalable, so it’s not clear to me why its applicability hasn’t been shown on more challenging datasets and models. I think this is an important point since the weight perturbations approaches from [Sharpness-Aware Minimization for Efficiently Improving Generalization](https://arxiv.org/abs/2010.01412) and [Adversarial weight perturbation helps robust generalization](https://arxiv.org/pdf/2004.05884.pdf) are very scalable (lead to only 2x overhead during training) and easily applicable to arbitrary deep networks.
- A discussion on some of the relevant prior works is missing, namely to [Sharpness-Aware Minimization for Efficiently Improving Generalization](https://arxiv.org/abs/2010.01412) (published at ICLR 2021) which is not cited and to [Adversarial weight perturbation helps robust generalization](https://arxiv.org/pdf/2004.05884.pdf) (published at NeurIPS 2020) which is cited but not discussed in detail. It is important to discuss them as the authors claim in the abstract: *“the first formal analysis”* and later in the text that: *“We note that previous methods use random weight perturbations to evaluate the generalization ability (e.g. sharpness), which we call a standard generalization setting. In contrast, we study a different problem setup: generalization of models trained under worst-case (adversarial) weight perturbation”*
Thus, I’d recommend the authors to discuss these references in detail and contrast their generalization bounds to the ones derived in this paper, potentially also including a numerical comparison between both.


**Other points**:
- Line 20: *“weight quantification”* -> *“weight quantization”*
- Line 22-23: *“weight perturbation sensitivity is also used as a metric to **evaluate** the generalization gap”* -- I think it’s worth being more precise here. What is shown in those papers is merely a **correlation**, and its exact relation to the generalization gap is not completely clear. So sharpness can’t be used to **evaluate** generalization gap.
- Line 285-287: “**will** suffer from a notable generalization gap” -> “**can** suffer from a notable generalization gap” -- as the derived upper bound is only an upper bound and on the exact quantity of interest.
- Line 308: citation is missing.
- Citation style: consider using \citet instead of \cite when citation is used as a part of a sentence (e.g., “[1] showed ...”)
- I think Theorem 3 can be put to the appendix as it doesn’t seem to give any new insights compared to Theorem 2.
- It’s worth numbering equations.


**Time Spent Reviewing:**

5

---

> ### Author Response · Authors · 2021-08-10
> **Response to Reviewer sPbL**
>
> We would like to first thank you for the sincere appreciation and the precious feedback! We are glad to receive encouraging feedback such as *“valuable contribution”* and *“Novel theory-driven loss function”* and would address the concerns raised in the review by points in the following.
>
> &nbsp;
> ### Regarding Relevant Works and Tightness of the Bound
> ___
>
> Although the bound in [R1] may seem similar since it is also derived in layer-wise fashion, we note that under our setting some intricacies are involved in deriving the worst-case error of the model under weight perturbation. Specifically, in deriving the weight perturbation bound, instead of directly using the matrix norm inequality all with respect to the $L_{\infty}$ norm, we adopted the notion of induced operator norm from $L_{1}$ and $L_{\infty}$ normed space (equation(9)-(10), in Appendix A.2), respectively, thereby avoiding the dimension dependence when straightly applying these inequalities. We will add this discussion in the revised version.
>
> Furthermore, we note that there are scenarios under which the worst-case error would occur by following when the related inequalities become equalities. For instance, it is possible to have constructed an input vector that happens to be in the direction of the matrix transformation (matrix norm in equation (7) - (8), Appendix A.2) while also equipped to be $1$-Lipschitz for the activation function (Lipschitz inequality in equation (6) - (7), Appendix A.2 ), making all the inequalities strictly equal. As an illustrating example, when one neural network with ReLU activation function possesses all positive parameters with its perturbation radius less than the actual value, entries in the input vector are all greater than zero, therefore tightening the overall bound. As a consequence, our derived bounds are tight unless one further restricts the space of possible neural network weights.
>
> &nbsp;
>
> [R1] Raghunathan, A., Steinhardt, J., and Liang, P. S. Semidefinite relaxations for certifying robustness to adversarial examples. In Advances in Neural Information Processing Systems, pp. 10899–10909, 2018b.
>
>
> &nbsp;
> ### Comparison of $L_{1}$/$L_{2}$ Regularization
> ___
>
> Thanks for your valuable insight in the constraint of weights regarding generalization.
> We would like to note that although these different norms all serve the purpose of reducing weight value, our regularization term acts in accordance with the proposed scenario (according to Theorem 4) and is the smallest of these techniques.
>
> Furthermore, the comparison to $L_{1}$ regularization was already presented in Appendix D.2, where we show our regularizer performs better.
>
> As an extension to our experiment conducted in D.2 regarding different forms of regularization, we’ve included the result of $L_{2}$ regularization below, which shows that these regularization methods can all serve to be one possible candidate in controlling the weights magnitude; however, using our loss the generalization gap is more reduced  and can better reflect model’s performance.
>
> &nbsp;
> #### Generalization Gap (%) / Test accuracy (with  $\epsilon= 0.01$)
> ___
>
> |  Coefficient $\mu$   |   0   | 0.002 | 0.004 | 0.006 | 0.008 |
> |:--------:|:-----:|:-----:|:-----:|:-----:|:-----:|
> |  $L_1$   |  6.36% / 84.2% |  6.09% / 84.8% |   Failed to Converge   |   Failed to Converge  |   Failed to Converge   |
> |  $L_2$   | 6.26% / 84.3% | 6.1% / 83.2%| 5.78% / 84.5%  | 5.6% / 78.8%|  5.5% / 78.5% |
> | our loss |  5.88% / 85.8% | 5.81% / 85%  | 5.51% / 87.8% |  5.41% / 83.4% | 5.1% / 85.1% |
>
> &nbsp;
> ### Experiments on Additional Dataset
> ___
>
> Following the reviewer’s suggestion, we’ve also included convolution-layer based model training on CIFAR using our proposed loss function in the table below. The model is primarily based on convolution layers and dense layers, with two convolution layers (32 filters of 5X5, and 64 filters of 5X5 in the second), three dense layers (128, 64, 10) and ReLU activation as prediction model.
>
> &nbsp;
> #### Convolutional Model Under PGD Weight Attack
> ___
>
> |              Perturbation Radius $\epsilon$               |   0    | 0.001  | 0.0015 | 0.0017 | 0.0019 | 0.0021 | 0.0023 | 0.0025 |
> |:-------------------------------------:|:------:|:------:|:------:|:------:|:------:|:------:|:------:|:------:|
> |         $\lambda=0, \mu=0$          | 58.73% | 13.08% | 10.15% | 10.01% |  10%   |  10%   |  10%   |  10%   |
> | $\lambda = 3.2*10^{-4}, \mu=10^{-4}$ |   51.1% | 33.9%  |23.16% |19.71% | 16.85%  | 14.25% |  12.52%      |11.58%        |
> | $\lambda=4.5*10^{-4},  \mu=10^{-4}$  | 47.28% | 36.82% | 27.78% | 26.19% | 21.6%  | 19.35% | 17.84% | 16.53% |
> |  $\lambda = 5*10^{-4}, \mu=10^{-4}$  | 42.71% | 38.68% | 30.2%  | 22.09% | 19.41% | 17.32% | 15.68% | 11.5%  |
> |         $\lambda=5.5*10^{-4}, \mu=10^{-4}$         | 42.79% | 32.9%  | 25.3%  | 22.77% | 20.78% | 18.88% | 18.26% | 17.1%  |
> |   $\lambda=6*10^{-4}, \mu=10^{-4}$   | 38.17% | 30.21% | 24.55% | 23.25% | 21.97% | 20.68% | 19.14% | 17.55% |
>
> As one can see, the standard model’s performance rapidly degrades in 5-folds when the perturbation radius is set to 0.001 while our model still retains half of its accuracy when compared to the no-perturbation setting ($\epsilon=0$). We will add the results to the revised version.
>
> &nbsp;
> ### Discussion of References
> ___
>
> Thank you for your precious suggestion and mentioning of these references. To make a more structured comparison, we will first present a table characterizing the key differences between our paper and then discuss them in detail.
>
> |  | Generalization Settings |Types of Robustness| Measure of Generalization Gap | Methods of Evaluating Maximum(Worst Case Loss)|
> | -------- | -------- | -------- | -------- | -------- |
> | Ours     | Robust Setting  | Against **Weight**|  $\mathbb{P}[\exists (w+\epsilon) s.t. \mathbb{1}(y \neq\arg\max [f_{w+\epsilon}(x)]_{y^{\prime}})] \leq \frac{1}{n} \sum \mathbb{1}(M(f_w(x),y) - \psi(f_w(x) \leq \gamma ) + \text{Gap} $ [See  Theorem 4]| Layer Propagation (Exact Bound)
> |Adversarial Weight Perturbation (2020)| Robust Setting |Against Input|$L_{D, \epsilon}(w+ \epsilon) \leq L_{S, \epsilon}(w+\epsilon) + \text{Gap}$ [See equation (12)]|Maximum value based on generated adversarial inputs (inexact bound)
> |Sharpness-Aware Minimization (2021)| Standard Setting|NA| $L_{D}(w) \leq \max_{\epsilon} L_{S}(w+\epsilon) + \text{Gap}$ [See Appendix A.1] |First-Order Taylor Expansion (approximation; inexact bound)
>
> In [R2], the authors extended the result of sharpness [R4] and integrated this concept as a part of the training process, namely the sharpness-aware minimization. In the paper, the author instead of directly solving the inner maximization loop, uses linear approximation to resolve the issue.
>
> Furthermore, the generalization bound given in [R2] considers only the standard setting, in which the generalization gap is measured between population risk and the sharpness-aware sample risk, differing from our adversarial setting where the generalization gap is contrast between both adversarial population and sample risk.
>
> On the other hand, the authors in [R3] propose to utilize weight perturbation to mitigate the widening robust generalization gap (based on **input** robustness). In [R3], the author trains the model by injecting the so-called “double perturbation” into the training process. We note that this is different from our approach of constructing adversarial robustness on weight space.
>
> Specifically, the weight perturbation in [R3] (second maximization step in equation (7), [R3])  is determined after every corresponding adversarial example is calculated, while our approach focuses on the worst-case weight perturbation given every sample data. Moreover, the generalization gap extends from [R4] where it is measured against the population risk and the sample risk along with perturbation direction sampled from zero mean spherical Gaussian distribution. On the contrary, our setting considers a generalization gap from both the adversarial population risk and sample risk.
>
> &nbsp;
>
> [R2]Foret, P., Kleiner, A., Mobahi, H., and Neyshabur, B. Sharpness-aware minimization for efficiently improving generalization. In International Conference on Learning Representations, 2021.
> [R3]D. Wu, S. Xia, and Y. Wang, “Adversarial weight perturbation helps robust generalization,” in NeurIPS, 2020.
> [R4]Neyshabur, B., S. Bhojanapalli, D. McAllester, et al. Exploring generalization in deep learning. In Advances in neural information processing systems, pages 5947–5956. 2017.
>
> &nbsp;
> ### On Other Points
> ___
>
> We thank you for pointing out corrections and presentation suggestions of the paper. We will follow your suggestions and update the paper accordingly.

---

> ### Author Response · Authors · 2021-08-31
> **Does our response address your concerns?**
>
> Dear Reviewer sPbL,
>
> Since the rolling discussion phase is closing soon, we would like to make sure your concerns have been fully addressed by our response.
> **In particular, we have clarified the generalization setting of ours as well as comparing our setting to other related works, discussed the tightness of our margin bound and further added two additional experiments:**
>
> - To compare different regularization against our proposed one;
> - To evaluate different model (convolution) on additional dataset.
>
> If there is anything we can do to convince you about the merits of this work, please let us know!

---

### Author Response · Authors · 2021-08-23
**Looking forward to reviewer's feedback**

Dear reviewers,

We would like to start by thanking all reviewers for the positive feedback and constructive comments given in the initial reviews. While the discussion deadline is approaching, we have not received any feedback based on our responses. We would like to use the interactive feature of OpenReview to engage the reviewers with the discussion. In particular, in our responses, we believe we have provided additional clarifications and new numerical results to fully address the reviewer's concerns. We hope our responses convince the reviewers about the merits of this work. If the reviewer has any other suggestions or comments, please don't hesitate to let us know!

Best Regards,

Authors

---

### Decision · Program_Chairs · 2021-09-27

**Decision:**

Accept (Poster)

**Comment:**

Even though the paper received mixed scores, all the reviewers agreed on the originality and the significance of the contributions of the paper. The authors’ extensive rebuttal letters later on clarified several concerns/confusions, especially in terms of lack of empirical evaluation. While the paper is borderline, I believe that most raised issues can be properly addressed **provided the authors implement all the required changes, which they addressed in their rebuttal**. Hence I am recommending a weak accept.